Article 

# Dynamical control enables the formation of demixed biomolecular condensates

Andrew Z. Lin [1,5], Kiersten M. Ruff [2,5], Furqan Dar[2,5], Ameya Jalihal[3], Matthew R. King[2], Jared M. Lalmansingh [2], Ammon E. Posey[2], Nadia A. Erkamp [2,4], Ian Seim [3], Amy S. Gladfelter[3] ✉ & Rohit V. Pappu [1] ✉

Cellular matter can be organized into compositionally distinct biomolecular condensates. For example, in *Ashbya gossypii*, the RNA-binding protein Whi3 forms distinct condensates with different RNA molecules. Using criteria derived from a physical framework for explaining how compositionally distinct condensates can form spontaneously via thermodynamic considerations, we find that condensates in vitro form mainly via heterotypic interactions in binary mixtures of Whi3 and RNA. However, within these condensates, RNA molecules become dynamically arrested. As a result, in ternary systems, simultaneous additions of Whi3 and pairs of distinct RNA molecules lead to well-mixed condensates, whereas delayed addition of an RNA component results in compositional distinctness. Therefore, compositional identities of condensates can be achieved via dynamical control, being driven, at least partially, by the dynamical arrest of RNA molecules. Finally, we show that synchronizing the production of different RNAs leads to more well-mixed, as opposed to compositionally distinct condensates in vivo.

Biomolecular condensates arise via spontaneous and driven phase transitions of complex mixtures of multivalent protein and RNA molecules[1–6]. A key challenge is understanding how condensates with shared and distinct macromolecular compositions can form and coexist as distinct functional entities[7] (Fig. 1a). A system that enables investigation of this problem is the filamentous fungus *Ashbya gossypii* where coexisting ribonucleoprotein condensates share the protein Whi3[8]. Specifically, the RNA encoding the cyclin *CLN3* forms a distinct condensate with Whi3 that does not colocalize with condensates formed by Whi3 and *BNI1* or *SPA2*[8], which are RNA molecules encoding proteins that regulate actin. These compositionally distinct condensates function in separate cellular processes[9–11] controlling nuclear division and cell polarity, respectively.

Rules underlying the formation of a single, well-mixed condensate versus compositionally distinct coexisting condensates can be predicted by mean-field theories where the relevant parameters are the variance of intermolecular interaction strengths and the numbers of components in multicomponent systems[12]. In ternary systems, such as mixtures of Whi3 with *CLN3* and *BNI1* in an implicit solvent, the relevant considerations are the interplay of solvent-mediated homotypic and heterotypic interactions[13] (Supplementary Fig. 1). Here, we first used lattice-based simulations, to extract thermodynamic criteria for evaluating whether condensates formed by binary Whi3-RNA mixtures are governed by purely heterotypic interactions or a blend of homotypic and heterotypic interactions[14]. This helps us discern the types of interactions that drive condensate formation. These criteria were then applied to analyze the phase behaviors of binary and ternary mixtures of the Whi3 protein with different RNA molecules in vitro. Our analyses implicate favorable heterotypic interactions as drivers of condensates in binary mixtures, leading us to account for and

[1]Division of Biology and Biomedical Sciences, Plant and Microbial Biosciences, Washington University in St. Louis, St. Louis, MO 63130, USA. [2]Department of Biomedical Engineering and Center for Biomolecular Condensates, James F. McKelvey School of Engineering, Washington University in St. Louis, St. Louis, MO 63130, USA. [3]Department of Cell Biology, Duke University, Durham, NC 27708, USA. [4]Yusuf Hamied Department of Chemistry, Centre for Misfolding Diseases, University of Cambridge, Lensfield Road, Cambridge CB2 1EW, UK. [5]These authors contributed equally: Andrew Z. Lin, Kiersten M. Ruff, Furqan Dar. ✉e-mail: amy.gladfelter@duke.edu; pappu@wustl.edu

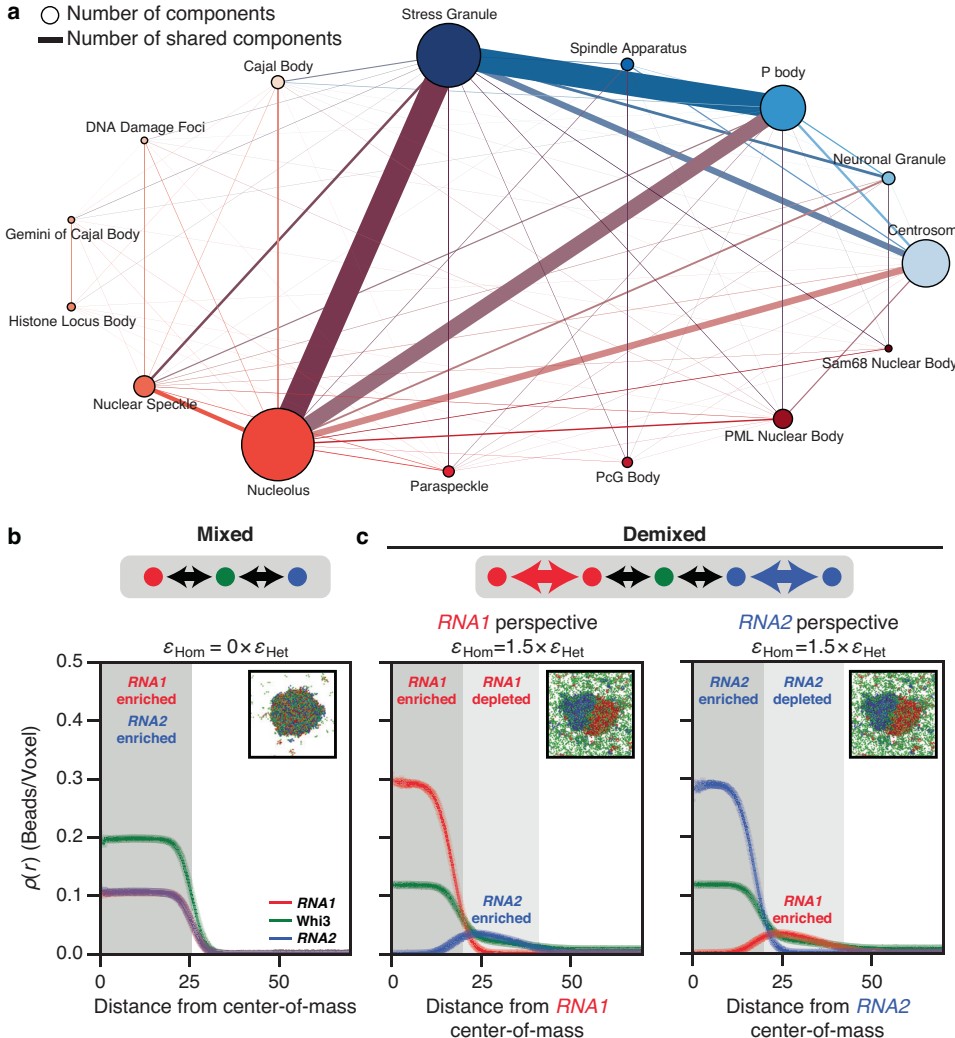

**Fig. 1 | Simulations show that the interplay between homotypic and hetero-typic interactions modulates the formation of mixed versus demixed phases. a** Graph of components in condensates in cells from *Homo sapiens* as extracted from DrLLPS[73]. Here, the node size correlates with the number of components within the condensate and the edge thickness quantifies the number of shared components between a pair of condensates that share an edge. Each nuclear con-densate is depicted as a node of a distinct size and reddish hue. Likewise, each cytoplasmic condensate is depicted as a node of a distinct size and blueish hue. Edges between nodes are shown in the color of the mixture of the colors of the two nodes. **b, c** Mean radial density profiles from three independent LaSSI lattice-based simulations of a ternary system designated as *RNA1* + *RNA2* + a Whi3 mimic (see "Methods"). Error bars denote the standard error of the mean. Black arrows denote

heterotypic interactions, whereas colored arrows denote homotypic interactions. **b** The system forms a well-mixed condensate when the RNAs engage in purely heterotypic interactions with the Whi3 mimic. Here, $\varepsilon_{\text{Het}} = -2k_BT$ refers to the strength of RNA interactions with the Whi3 mimic and $\varepsilon_{\text{Hom}} = 0 \times \varepsilon_{\text{Het}}$ implies that homotypic interactions are zeroed out. **c** In contrast, demixed condensates are formed when the RNAs have strong homotypic interactions as well, where the strength of homotypic interactions is set to be 1.5 times that of heterotypic inter-actions, i.e., $\varepsilon_{\text{Hom}} = 1.5 \times \varepsilon_{\text{Het}}$. The right two plots show results from the same simulations. In the middle plot, the macromolecular densities are quantified from the perspective of the center-of-mass of *RNA1* whereas in the plot on the right, the density profiles are plotted from the perspective of the center-of-mass of *RNA2*. Source data are provided as a Source Data file.

demonstrate how dynamical considerations are the main drivers of compositionally distinct condensates formed by Whi3 with different RNA molecules. We show that dynamical control can also direct the formation of compositionally distinct Whi3 condensates in cells. This work provides a molecular mechanism for explaining how con-densates that share some molecular components can nevertheless coexist and retain distinct compositional identities.

## Results

### Interactions that produce compositionally distinct condensates
We used computations to ascertain the types of interactions in ternary systems that can lead to the spontaneous formation of two distinct coexisting dense phases with one shared component. We used explicit lattice-based simulations as opposed to mean-field models because

they allow for proof-of-concept titrations of different variables and require fewer simplifying assumptions. We performed lattice-based coarse-grained simulations[15,16] of a model system in which two high-valency linear polymers, designated as *RNA1* and *RNA2*, interact with the same low-valency linear polymer (a Whi3 mimic) with varying types of interactions[17]. Here, valency refers to the numbers of cohesive motifs, which we refer to as stickers in accord with the nomenclature used in the field of associative polymers[18–20]. Stickers can engage in complementary homotypic or heterotypic interactions[21]. Note that the models we use are phenomenological as opposed to being faithful representations of the Whi3 protein and RNA molecules.

When the interactions of *RNA1* and *RNA2* mimics with the Whi3 mimic are the same in strength and purely heterotypic in nature, the dense phase comprises all three macromolecules (Fig. 1b). In contrast,

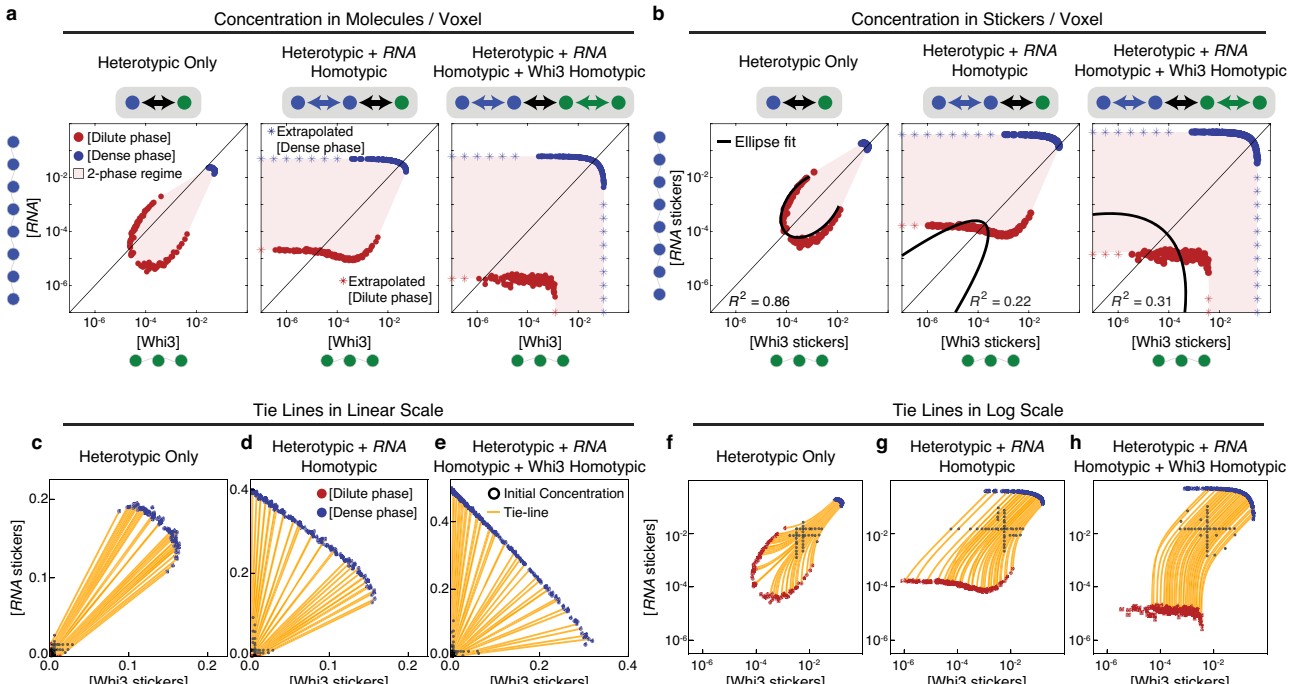

**Fig. 2 | Phase boundaries for two-component systems can be used to infer relative interaction strengths of homotypic and heterotypic interactions.** **a** Phase boundaries computed using LaSSI simulations for a system with RNA and Whi3 mimics when interactions are purely heterotypic (left), a system where equivalent homotypic interactions are included among RNA molecules (center), and a system where equivalent homotypic interactions are included for both the RNA and Whi3 mimics (right). Here, the RNA and Whi3 mimics have sticker valencies of eight and three, respectively. **b** The phase boundaries in panel (**a**) are recast in terms of concentrations of stickers. Black lines denote elliptical fits and the $R^2$ values quantify the goodness of the fit. **c**–**h** Example tie lines for the phase boundaries in panel (**b**). From the computed phase boundaries, we show a subset of starting concentrations and the corresponding tie lines. Panels (**c**–**e**) are linear plots, while panels (**f**–**h**) are log-log plots. For each phase diagram, two independent replicas were performed. Source data are provided as a Source Data file.

we observe distinct dense phases when *RNA1* and *RNA2* have stronger homotypic interactions than heterotypic interactions with the Whi3 mimic (Fig. 1c). In this scenario, one dense phase is enriched in *RNA1* and the other in *RNA2*; the Whi3 mimic is equivalently enriched in both (Fig. 1c, Supplementary Figs. 2 and 3). We also investigated the possibility of realizing spontaneous demixing through other interaction modes[22]. Selective repulsion between different RNA species, whereby *RNA1* and *RNA2* repel one another while being agnostic toward themselves can lead to spontaneous demixing depending on the strength of the repulsion vis-à-vis the favorable heterotypic interactions (Supplementary Figs. 4 and 5). The agnosticism of homotypic RNA interactions could arise from a screening of attractions by repulsions, as would be the case in a Flory melt or if the solvent is an effective theta solvent for the specific set of interactions[17,23]. Another possibility that was considered was asymmetric heterotypic interactions with the shared component[22,24]. Here, the interactions between the Whi3 mimic and the *RNA1* and *RNA2* mimics are different from one another. In the context of our model, strongly asymmetric heterotypic interactions lead to minimal demixing (Supplementary Figs. 6 and 7).

The simulation results, taken together with published theoretical and computational studies[13,16,22,24,25], show that strong homotypic interactions between components that are not shared across condensates provide one modality, albeit not the only one, for forming compositionally distinct condensates. Our analysis highlights the importance of assessing the relative contributions of homotypic versus heterotypic interactions to the phase behaviors of Whi3 with different RNA molecules. To interpret in vitro measurements, we first used computations to demonstrate how the relative contributions of homotypic and heterotypic interactions can be extracted from phase diagrams of two-component systems. The question we answer is as follows: given a measured phase diagram for a Whi3-RNA mixture, can

we determine whether heterotypic Whi3-RNA interactions or homotypic protein-protein and RNA-RNA interactions are the dominant drivers of the observed phase behaviors?

## Assessing contributions of homotypic versus heterotypic interactions

We analyzed phase boundaries generated using LaSSI simulations of two-component systems[4,15]. For these simulations, we used a model system comprising low- and high-valency polymers, representing Whi3 mimics and RNA mimics, respectively. Specifically, we consider the differences between two-component systems defined by purely heterotypic interactions versus one where the longer polymer also engages in equivalent homotypic interactions and another where both polymers have equivalent homotypic interactions.

Figure 2a shows the phase boundaries plotted in terms of the concentrations of molecules for the three systems studied. For the system defined by purely favorable heterotypic interactions the phase boundary forms a closed loop, and the dilute arm is convex with respect to the diagonal[2,15,26,27]. Unfavorable heterotypic interactions, such as electrostatic interactions, will lead to local concavities in the phase boundaries. In contrast, when homotypic interactions of the RNA mimic are equivalent to heterotypic interactions with the Whi3 mimic, the RNA mimic can undergo phase separation on its own. At low concentrations of the Whi3 mimic, the phase boundary broadens and intersects horizontally with the axis along which the concentration of the RNA mimic is titrated. The intersection indicates the intrinsic apparent saturation concentration $c_{sat}$ of the RNA mimic in the absence of the Whi3 mimic (Fig. 2a). Similarly when both the RNA and Whi3 mimics have homotypic interactions, they are capable of phase separating individually and the two-phase regime intersects with both axes.

For phase separation driven by purely heterotypic interactions, the phase boundary should be symmetrical about the diagonal[2,5,15,26–28]. While this is true of molecules with equivalent numbers of stickers, it should also be true if the titrations along each of the axes quantify the concentrations of stickers rather than molecules[27]. For the case of purely heterotypic interactions, phase separation should be most favored when sticker concentrations are balanced between the two molecules. Figure 2b and Supplementary Fig. 8 show a remapping of phase boundaries in terms of sticker concentrations. Although the remapping does not use prior knowledge of the number of stickers, this knowledge is useful because it helps us test the accuracy of the procedure developed to remap phase diagrams in terms of stickers whose valence is extracted post facto. We quantified the degree of symmetry about the diagonal by fitting an ellipse to the dilute arm of the measured and mirrored phase boundary (see "Methods"). The addition of homotypic interactions breaks the symmetry about the diagonal and leads to poorer fits of a single ellipse (Fig. 2b and Supplementary Fig. 8). Therefore, remapping phase boundaries in terms of concentrations of stickers, and quantifying the degree to which the dilute concentration arm can be described by a single conic section such as an ellipse, helps with discerning the interplay between heterotypic and homotypic interactions.

Additional insight into the balance of homotypic and heterotypic interactions comes from the slopes of tie lines that connect the concentrations of coexisting phases[6]. In a plane defined by the concentrations of Whi3 and RNA mimics, the slopes of tie lines provide insights regarding components that are preferentially excluded or included within dense versus dilute phases[6,29]. The slopes of tie lines can also provide insights regarding the interplay between heterotypic versus homotypic interactions, providing the interactions are non-equivalent[14].

In the absence of hidden components within the solvent that differentially partition between dense and dilute phases, the tie lines plotted on the plane defined by two components will be straight lines[30] (Fig. 2c–e). This is true if phase separation leads to two coexisting phases. Tie lines become tie simplexes for more than two coexisting phases. For a two-component system, defined by purely heterotypic interactions, the slopes of tie lines are governed by the stoichiometric ratios of the concentration of stickers in the Whi3 versus RNA mimics or vice versa (Fig. 2c). The addition of homotypic interactions tilts the slopes toward the axis corresponding to the component that adds homotypic interactions to extant heterotypic interactions (Fig. 2d). As with the case of purely heterotypic interactions, the equivalence of homotypic and heterotypic interactions among the two components will again lead to a pure stoichiometry dependence to the slopes of tie lines (Fig. 2e). The key difference between the scenario defined by purely heterotypic interactions versus equivalent heterotypic and homotypic interactions is the width of the two-phase regime[14]. This is seen by comparing Fig. 2c–e. The results in Fig. 2b are shown as log-log plots whereas the tie lines in Fig. 2c–e are shown on a linear scale. Recasting the phase boundaries as log-log plots, as shown in Fig. 2b, leads to curved tie lines (Fig. 2f–h), and this is true for all interaction modes. Note that the solvent is implicit in our simulations and hence, there are no hidden components[30].

The analyses summarized in Fig. 2c–h are relevant given recent qualitative[5,15] and quantitative interpretations of the slopes of tie lines[28,29]. For example, Qian et al.[29] have proposed that the shapes of phase boundaries can be uninformative regarding the contributions of homotypic versus heterotypic interactions when compared to the slopes of tie lines. Our analysis, summarized in Fig. 2b, and the recent work of Farag et al.[14], suggest otherwise. Comparisons of the results in Fig. 2c and e show that the slopes of tie lines cannot be relied upon without a comparative assessment of the widths of the two-phase regimes. Also essential is the shape of the dilute arm that helps distinguish expectations for purely heterotypic interactions versus the case defined by a blend of equivalent heterotypic and homotypic

interactions[14]. Therefore, the curved tie lines in a log-log plot and/or the shapes of phase boundaries in a log-log plot combined with an assessment of whether the phase boundary can be symmetrized about the 1:1 line can be informative regarding the relative contributions of homotypic versus heterotypic interactions. We use this as a numerical strategy to analyze phase boundaries measured in vitro for Whi3-RNA mixtures.

## Heterotypic interactions dominate in Whi3-RNA mixtures

Next, we applied insights from our computations to interpret measured phase boundaries of distinct binary mixtures in vitro. Figure 3a shows the measured phase boundaries for Whi3 with each of three different cognate RNA molecules, viz., *CLN3*, *BNI1*, and *SPA2*. Qualitatively, the phase boundaries do not show signatures of dominant homotypic interactions of Whi3 or any of the RNAs given the convex shapes of the dilute arms. To test this hypothesis, we determined the degree to which measured phase boundaries could be symmetrized about the diagonal by assessing the fit of a single ellipse.

The measured phase boundaries were remapped in terms of sticker concentrations (Fig. 3b). Stickers can enable homotypic and heterotypic interactions. For our analysis of measured phase boundaries, the null hypothesis is that the phase behavior is driven primarily by heterotypic interactions. The glutamine-rich region and the RRM of Whi3 are essential for driving phase separation[31]. The RRM engages in site-specific interactions with cognate RNA molecules. Accordingly, we set the valence of RNA-binding stickers on Whi3 to be one and titrated the valence of stickers in RNA molecules as a free parameter. We pick the valence that maximizes the symmetry of the dilute arm about the diagonal. By doing so, we extract the relative sticker valence of the RNA with respect to Whi3 which we term the apparent sticker valence (Supplementary Fig. 9).

Given the importance of the apparent sticker valence, we describe the method by which this parameter was extracted for each binary Whi3-RNA system. First, we extracted n points along the lower boundary of the two-phase regime $\mathbf{x_P} = (x_{P1}, x_{P2}, \ldots x_{Pn})$ and $\mathbf{y_P} = (y_{P1}, y_{P2}, \ldots y_{Pn})$, and m points along the upper boundary of the one-phase regime $\mathbf{x_N} = (x_{N1}, x_{N2}, \ldots x_{Nm})$ and $\mathbf{y_N} = (y_{N1}, y_{N2}, \ldots y_{Nm})$. Here, the subscript N and P represent points in the one- (**N**o phase separation) and two-phase (**P**hase separation) regimes, respectively. As our null hypothesis, we assume that the system is dominated by heterotypic interactions. Using this assumption, we rescaled the RNA sticker concentrations ($\mathbf{y_P}$ and $\mathbf{y_N}$) by varying a scaling factor, $s$, to find the highest degree of symmetry. The optimal $s$ should generate the lowest overlap between the area defined by ($\mathbf{x_N}$, $s\mathbf{y_N}$) and ([$\mathbf{x_P}$ $s\mathbf{y_P}$], [$s\mathbf{y_P}$ $\mathbf{x_P}$]). If heterotypic interactions dominate, then there should be minimal overlap between the area defined by the upper boundary of the one-phase regime and the area defined by the lower boundary of the two-phase regime plus its mirrored data. We determined the overlap area as described in "Methods". The apparent sticker valence of the RNA was taken to be the value of $s$ that yields the minimum overlap area. Lastly, for the value of $s$ that corresponds to the minimum overlap we fit a closed loop to ([$\mathbf{x_P}$ $s\mathbf{y_P}$], [$s\mathbf{y_P}$ $\mathbf{x_P}$]) using the fitting method of Szpak et al.[32], and extracted the $R^2$ of the fit.

The measured phase boundaries, remapped onto sticker concentrations, can be fit by a single closed loop with $R^2$ values of 0.88-0.92 (Figs. 3b and 4a). This remapping suggests that heterotypic interactions are the main drivers of Whi3-RNA phase separation. As shown in Fig. 2, the results of symmetrization shown in the panels of Fig. 3b would have failed if homotypic interactions were on an equal footing with or dominated heterotypic interactions. That heterotypic interactions are essential for condensate formation at 150 mM KCl is further emphasized by the fact that in the absence of Whi3, the condensation of RNA alone requires KCl concentrations that are in the molar regime, with the precise values being RNA-dependent (Supplementary Fig. 10). Our findings do not imply the absence of homotypic interactions. Instead, they suggest that, on balance, condensate

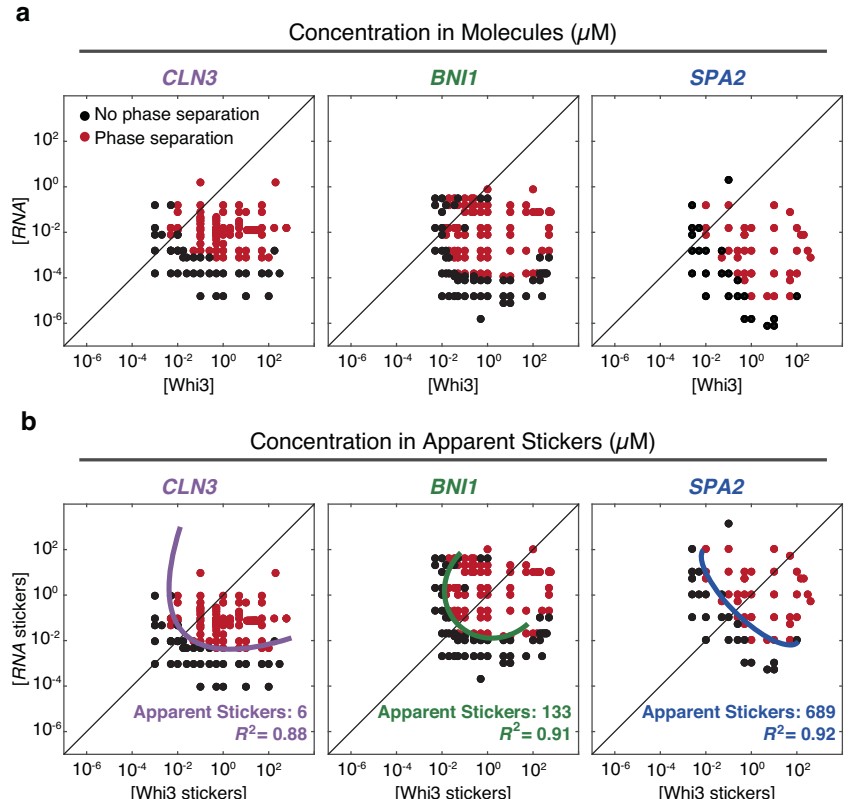

**Fig. 3 | Measured phase boundaries for Whi3-RNA mixtures. a** Phase boundaries, measured in vitro, of Whi3 with *CLN3* (left), *BNI1* (center), and *SPA2* (right) determined by confocal microscopy. Here, the phase boundaries are depicted in terms of concentrations of macromolecules. **b** The data in each of the panels of row (**a**) are recast in terms of concentrations of stickers (see "Methods"). The data were fit to ellipses and the fits are shown in solid color lines. The legend summarizes the apparent sticker valence and the $R^2$ values that quantify the goodness of the fit. Source data are provided as a Source Data file.

formation is driven by heterotypic interactions that dominate over homotypic protein-protein and RNA-RNA interactions[14].

### RNA sequences are distinguishable by their apparent sticker valence

The in vitro measurements of binary Whi3-RNA mixtures provide a detailed mapping of the low-concentration arms of Whi3-*CLN3*, Whi3-*BNI1*, and Whi3-*SPA2* phase boundaries. The results of symmetrization and testing the null hypothesis that phase boundaries are dominated by heterotypic interactions are summarized in terms of the consensus closed loops used to fit each of the measured phase boundaries (Fig. 4a, Supplementary Fig. 11). Comparison of the boundaries suggests a hierarchy of driving forces for phase separation in Whi3-RNA mixtures. Along the diagonal, the driving forces are strongest in Whi3-*CLN3* mixtures and weakest in Whi3-*SPA2* mixtures, with Whi3-*BNI1* mixtures being defined by driving forces of intermediate strengths (Supplementary Fig. 11). This is noteworthy because *CLN3* is the shortest of the three RNAs and *SPA2* is the longest.

Whi3 binds the consensus sequence UGCAU which we refer to as the cognate-binding site[33]. The numbers of cognate-binding sites for Whi3 are 5 each for *CLN3* and *BNI1* and 11 for *SPA2*. The symmetrization procedure, which accounts for the overall shape of the measured phase boundary, recasts the phase boundaries in terms of concentrations of stickers rather than molecules. This analysis leads to the inference that *CLN3*, *BNI1*, and *SPA2* have apparent sticker valences of 6133, and 689, respectively (Fig. 4b). The implication is that there are additional non-cognate stickers within each of the RNA molecules. The boxplots in Fig. 4b correspond to the range in apparent valence values if we allow for a five percent change in the minimum area.

The numbers of non-cognate sites increase with length, and the ratio of non-cognate to cognate stickers is 0.2, 25.6, and 61.6 for *CLN3*, *BNI1*, and *SPA2*, respectively. One plausible conjecture regarding the nature of the non-cognate stickers is that they signify the contributions of phosphate groups to non-specific, complementary electrostatic interactions between Whi3 and RNA molecules. If this were the only contribution to condensation in Whi3-RNA mixtures, then the expectation would be of a linear scaling between the apparent valence and RNA length with a slope on a log-log plot being close to unity. Instead, the slope of the log-log plot of apparent sticker valence versus RNA length is 2.4 rather than unity (Fig. 4c). This suggests that there are Whi3-RNA interactions that involve non-cognate sites that contribute as auxiliary stickers and the numbers of these sites do not scale trivially with polynucleotide length, although there is a positive correlation with length.

If there are non-cognate sites within longer RNA molecules, then disruption of cognate sites should still preserve phase separation. To test for this, we measured in vitro phase boundaries for the two-component system comprising Whi3 and a *BNI1* mutant (*BNI1mut*) in which the cognate-binding sites were disrupted by scrambling the consensus sequence. Disruption of cognate sites does not abrogate phase separation. However, there are two prominent changes to the shape of the phase boundary. Heterotypic interactions are likely to be a blend of favorable and unfavorable contributions. Scrambling the cognate sites alters the shape of the dilute arm. Specifically, we note a switch from being locally convex to locally concave, implying a more prominent contribution from unfavorable heterotypic interactions at low concentrations of Whi3 and the mutant RNA when compared to the wild-type *BNI1* (Fig. 4d). Further, at low Whi3 concentrations, a higher concentration of *BNI1mut* compared to

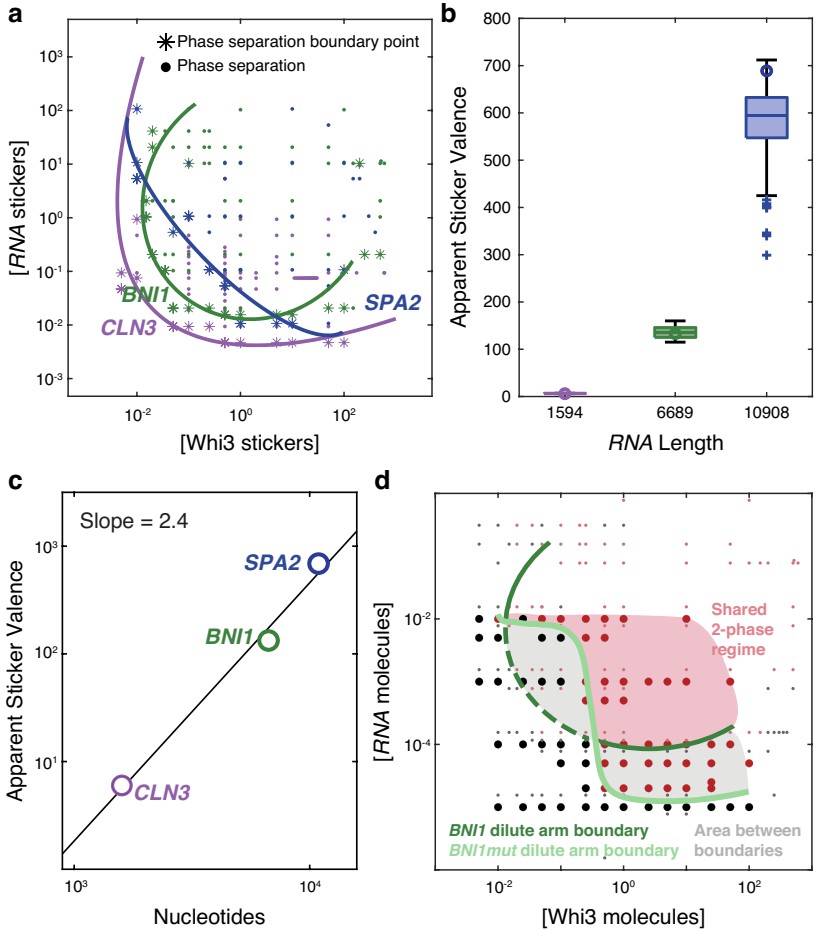

**Fig. 4 | Non-cognate stickers contribute to Whi3-RNA phase separation. a** The derived phase boundaries, recast in terms of concentrations of protein and RNA stickers are shown for each of the three Whi3-RNA systems. Here, colored points represent concentrations at which phase separation was observed by confocal microscopy. Asterisks denote the points that were used to generate consensus elliptical boundaries for each of the three systems. **b** Plot of the inferred apparent valence of stickers for each of the RNA molecules (see "Methods"). Circles denote the apparent sticker valence that corresponds to the minimum overlap between the one- and two-phase regimes. Boxplots are shown for all apparent sticker valences within a 5% change of this minimum (where $n = 244$, and 143 for *CLN3*, *BNI1*, and *SPA2*, respectively). The median is shown as a colored horizontal line, whiskers show the 1.5 interquartile range, outliers are shown as pluses, and the bottom and top of each box are the 25th and 75th percentiles, respectively. **c** Log-

log plot of nucleotides versus apparent sticker valence for *CLN3*, *BNI1*, and *SPA2* RNAs. **d** Measured phase boundaries for Whi3-*BNI1* and Whi3-*BNI1mut* mixtures. Here, black, and dark red points denote points in the one- versus two-phase regime for the Whi3-*BNI1mut* system, respectively, whereas gray and light red points denote points within the one- versus two-phase regime for the Whi3-*BNI1* system, respectively. A dark green ellipse is fit to the measured phase boundary for the Whi3-*BNI1* system, whereas the light curve denotes the low concentration arm of the measured phase boundary of the Whi3-*BNI1mut* system. Scrambling the cognate Whi3-binding sites on *BNI1* eliminates phase separation in the low concentration regime (sub-micromolar, gray area) for Whi3 and *BNI1mut*. All concentrations for in vitro measurements are in units of μM. Source data are provided as a Source Data file.

*BNI1* is needed for phase separation, whereas at high Whi3 concentrations, lower concentrations of the mutant RNA are sufficient to drive phase separation. Thus, while non-cognate sites can drive phase separation, their interactions are weaker, across the concentration ranges investigated, than those of cognate-binding sites of *BNI1*.

## Whi3-RNA condensates are resistant to increases in monovalent salt

Complex coacervation is a form of phase separation that is driven mainly by complementary electrostatic interactions among oppositely charged macromolecules[34–37]. Condensates driven by complementary electrostatic interactions should show salting-in behavior, implying that they should dissolve as the concentration of salt increases. To test whether non-cognate interactions are purely electrostatic in nature, we measured the salting-in transitions of Whi3-RNA condensates. The results are summarized in Fig. 5. At low salt, ca. 50 mM KCl, we observed co-condensation of Whi3 and RNA molecules in binary mixtures of Whi3 with *BNI1*, *BNI1mut*, and *CLN3*. Co-condensation persists for higher salt

concentrations, and we do not observe the salting-in behaviors one would expect for complex coacervation. However, at high salt concentrations that are in the molar regime, we observe a weakening of Whi3 signals in condensates. Notably, despite a decrease in dense phase Whi3 concentration, the condensates persist and remain resistant to increased salt concentrations. Further, and surprisingly, at high salt concentrations, the Whi3 molecules released from Whi3-RNA condensates can form protein-only condensates. This points to hitherto unknown reentrant behaviors[38,39] whereby Whi3 drives protein-only condensation via an apparent salting-out transition.

The work of Langdon et al. pointed to the presence of complementary sites among RNA molecules[8]. Concordant with their analyses, we find that there are different numbers of complementary sites among the different RNA molecules (Supplementary Fig. 12). These sites are the likely contributors to maintaining salt-resistant, RNA-rich condensates. Overall, our results suggest that heterotypic interactions between Whi3 and RNA molecules are the main drivers of phase separation that give rise to Whi3-RNA condensates in binary mixtures

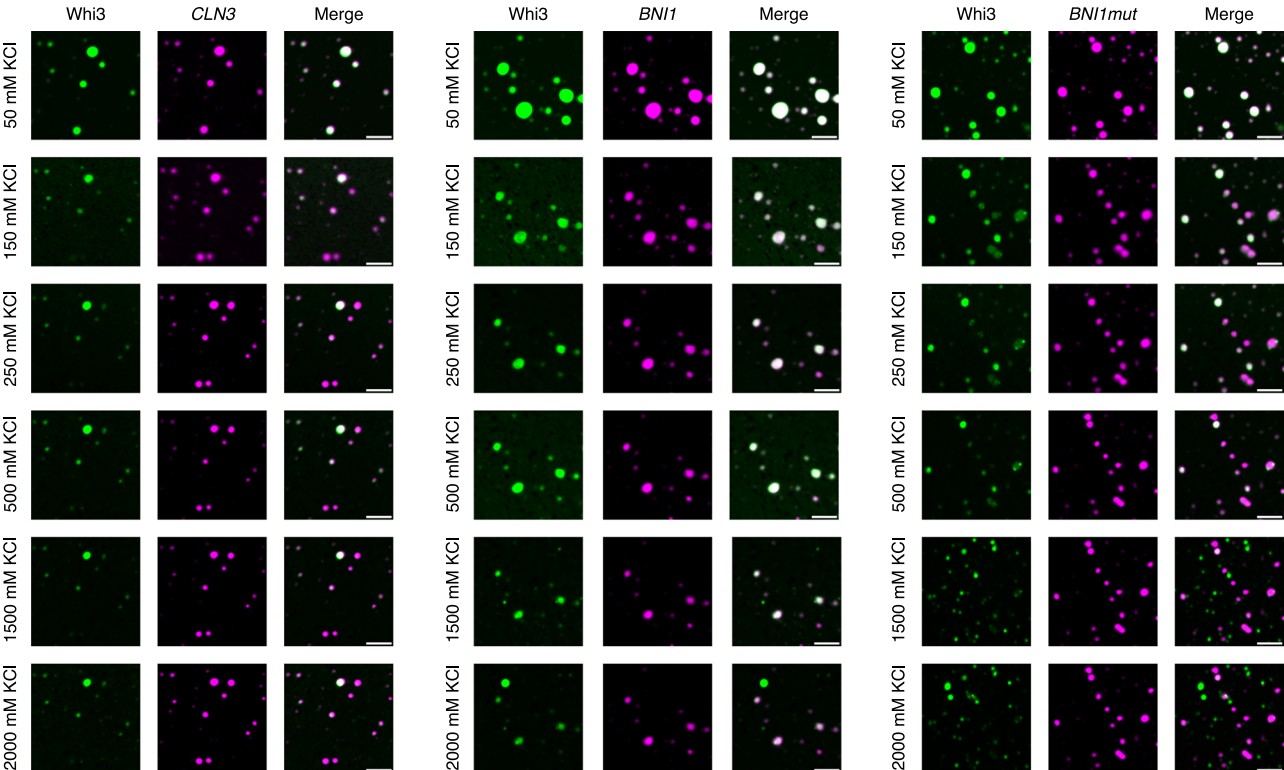

**Fig. 5 | Whi3-RNA condensates are salt-resistant, although the levels of Whi3 can decrease as salt concentrations increase.** Whi3-RNA condensates were formed at 50 mM KCl and allowed to incubate for 5 min at room temperature in a 384-well plate with a #1.5 glass coverslip bottom. Differing amounts of KCl were added to each sample and mixed gently by pipetting to obtain the next successive salt concentration. Samples were allowed to incubate for 5 min after high salt addition before imaging. This process was performed twice, with three independent fields of view for each sample. The condensates were imaged while maintaining the fields of view, and the protein (green) and RNA channels (pink) were imaged separately. Purely pink condensates in the merged columns indicate a weakening or loss of Whi3 signals due to the loss of proteins from the condensates. Purely green condensates in the merged column signify protein-only condensates. Scale bar represents 5 μm for all images.

at physiologically relevant salt concentrations. Heterotypic interactions involve Whi3 and cognate as well as non-cognate sites on RNA molecules. Homotypic RNA interactions within condensates appear to enable the stabilization of salt-resistant, RNA-rich, protein-deficient condensates.

### RNA molecules are immobile in Whi3-RNA condensates

Next, we probed the internal dynamics of protein and RNA molecules in condensates formed by Whi3 and four different RNA species. For this, we measured fluorescence recovery after photobleaching (FRAP) of Whi3 and RNA in each of the Whi3-RNA condensates. Although up to ~20% of the Whi3 fluorescence can be recovered after photobleaching (Fig. 6a, c), the RNA molecules are essentially immobile, showing undetectable FRAP in all the condensates formed by binary mixtures (Fig. 6b, c). The recovery of Whi3 fluorescence is highest in Whi3-BNI1mut condensates. When compared to BNI1, this implies that the presence of cognate-binding sites decreases Whi3 mobility. When comparing Whi3-CLN3 and Whi3-BNI1 condensates, we find that despite BNI1 being longer, and the two sequences featuring essentially similar numbers of cognate sites, co-condensation with the longer RNA enables higher mobility of Whi3 molecules. Finally, we find that the extents of recovery are similar for BNI1 and SPA2 even though the latter is almost twice as long as BNI1.

### Compositionally distinct condensates form under dynamical control

Within condensates, RNA molecules become immobile, and we hypothesize that this dynamical arrest might be a source of demixing behavior. The dynamics of phase separation in multicomponent mixtures will be governed by the interplay between two timescales. The first is molecular transport, such as diffusion of molecules, and the second is the timescales associated with the making and breaking of physical crosslinks among molecules[40–42]. These dynamical processes will be influenced by the stoichiometry of stickers that enable complementary homotypic and heterotypic interactions (Supplementary Fig. 12)[42]. Asymmetries of timescales, which we refer to as asynchronies that are caused either by long-lived crosslinks impacting molecular transport[40] or significant asymmetries in molecular mobilities[43] should give rise to dynamically controlled phase separation[40].

To test whether dynamical factors lead to the demixing of Whi3-RNA condensates, we analyzed the extent of colocalization within in vitro condensates formed in ternary systems comprising two types of RNAs and the Whi3 protein. The concentrations of Whi3 (5 μM) and different RNA molecules (5 nM) were chosen to ensure that these lie within the two-phase regimes in binary mixtures (Fig. 3a). The condensates were formed using two different methods. The first method, termed *delayed* involves mixing Whi3 with one of its cognate RNAs, waiting for 4 h, and then adding in the second RNA (Fig. 7a). The second method, termed *simultaneous*, involved adding the two RNAs simultaneously to Whi3 (Fig. 7b). To quantify the extent of demixing, colocalization between RNA channels was determined on a pixel-by-pixel basis and quantified using a Pearson correlation coefficient (Pearson's r).

The extent of colocalization of *CLN3* to pre-formed Whi3-BNI1 condensates was low when condensates in ternary mixtures were prepared using the *delayed* approach (Fig. 7c, i). In contrast, *simultaneous* mixing led to a higher degree of colocalization of *CLN3* and *BNI1* in condensates with Whi3 (Fig. 7d, i). This difference in colocalization

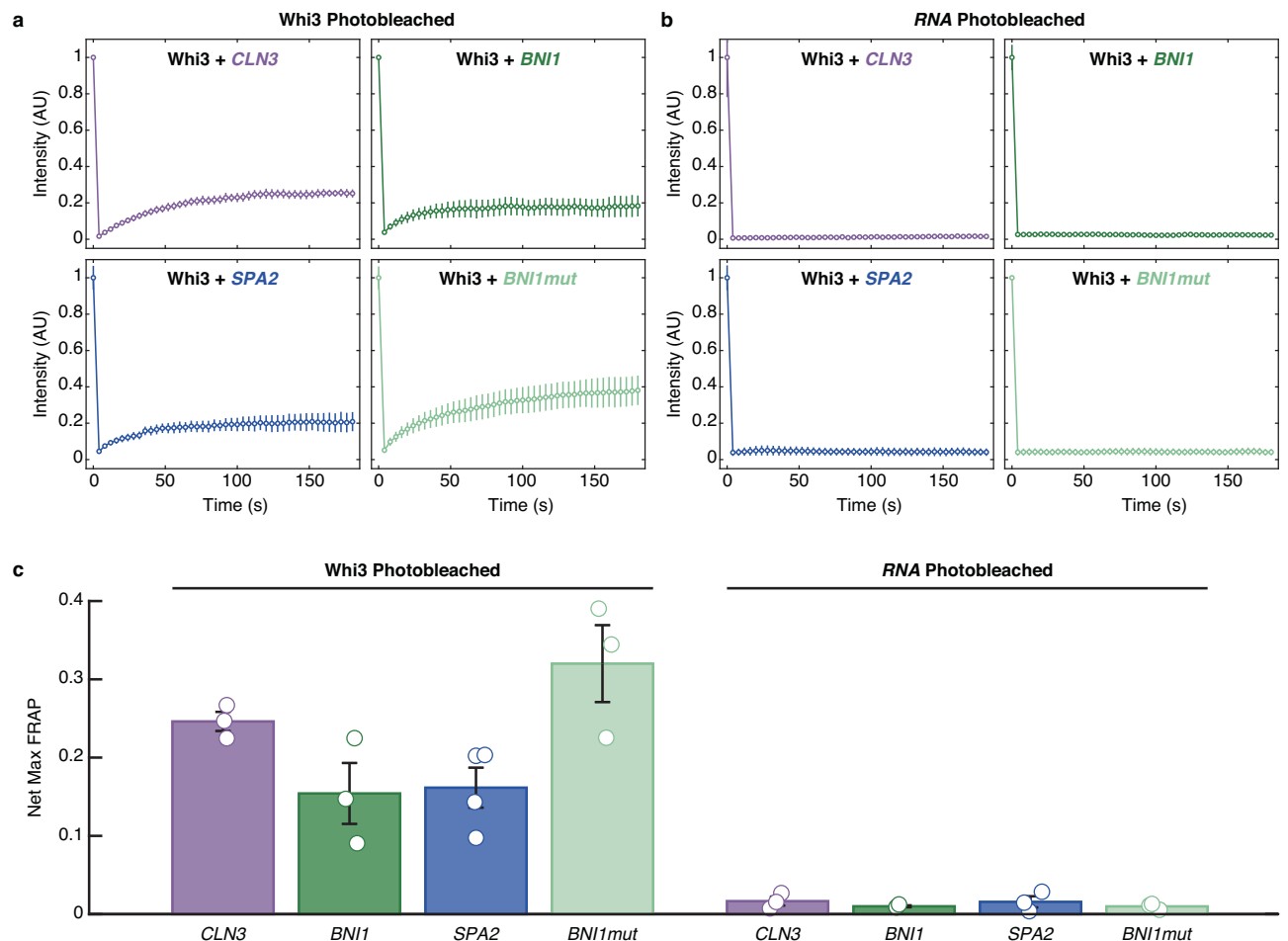

**Fig. 6 | FRAP measurements highlight the relative immobility of RNA molecules in Whi3-RNA condensates. a** The collection of four plots that make up this composite panel shows roughly 20% recovery of Whi3 protein fluorescence depending on the Whi3-RNA condensate that is being interrogated. As summarized in panel (**c**), recovery is the largest for Whi3 within Whi3-*BNI1mut* condensates. **b** Across all Whi3-RNA condensates, the recovery of RNA fluorescence is essentially negligible.

**c** Net maximal recovery in normalized fluorescence values from FRAP traces for Whi3 and RNA in binary Whi3-RNA systems. **a**–**c** Error bars denote the standard error of mean across three independent experiments, except for the Whi3 FRAP with *SPA2* which had four independent experiments. Source data are provided as a Source Data file.

was consistent for the Whi3, *BNI1*, and *SPA2* ternary mixture. However, we observed that *SPA2* had higher colocalization than *CLN3* in the *delayed* approach, consistent with the tendency of *SPA2* to colocalize with *BNI1* in vivo[8] (Fig. 7e, f, i). Overall, the extents of colocalization of RNA molecules were highest in the *simultaneous* mode of condensate preparation. We performed finer titrations of the temporal delay in adding the second RNA component. We find that even a 30-min delay in addition can lead to diminished colocalization of the RNAs (Supplementary Fig. 13), whereas the degree of colocalization in the *simultaneous* mode is high, at all the time points studied (Supplementary Fig. 14).

To test the effect of cognate-binding sites, we quantified the extent of colocalization achieved in ternary mixtures of Whi3, *BNI1mut*, and *CLN3*. Scrambling the cognate-binding sites for Whi3 increased the extent of colocalization in the *delayed* mode (Fig. 7g, i). This result suggests that cognate-binding sites contribute more significantly to long-lived Whi3-RNA crosslinks and dynamical arrest of RNA molecules when compared to non-cognate sites.

If dynamical arrest due to metastable traps is a driver of demixing, then *delayed* addition of the same RNA species to a pre-formed Whi3-RNA condensate should also result in demixing. Therefore, we labeled the same RNA molecules with two different dyes (Fig. 8a, b). As expected, we observed reduced colocalization between the *CLN3* molecules labeled with different dyes in the *delayed* mode (Fig. 8c, i). In

the *simultaneous* mode, we observed a high degree of colocalization between the *CLN3* molecules labeled with different dyes (Fig. 8d, i). Similar results were observed for Whi3-*BNI1* and Whi3-*BNI1mut* mixtures, suggesting that demixing driven by metastable traps might be a general phenomenon (Fig. 8e–i). However, in general and in accord with the work of Langdon et al., we observed that the extent of demixing in the delayed mode was strongest when *CLN3* happened to be one of the RNA molecules.

To explain the totality of our results regarding dynamically controlled demixing of Whi3-RNA condensates, we propose that there are at least three different factors. The first is long-lived physical crosslinks formed in Whi3-RNA condensates. These crosslinks can be protein-RNA or RNA-RNA interactions. We refer to this as crosslinking-limited dynamical arrest in line with observations made by Ranganathan and Shakhnovich[40] as well as Lee[44]. The second is the number and accessibility of complementary sites[45] within the condensate for the third component (Supplementary Fig. 12). We find that *CLN3* has the smallest number of complementary sites for homotypic or heterotypic RNA-RNA interactions. This contrasts with the number of complementary sites for complementary interactions among *BNI1mut* and *BNI1* (Supplementary Fig. 12). The implication is that *CLN3* has low complementary valence, and this is in contrast to the other RNA sequences. This limitation of complementary sites limits its ability to be recruited to preexisting condensates based on either heterotypic or

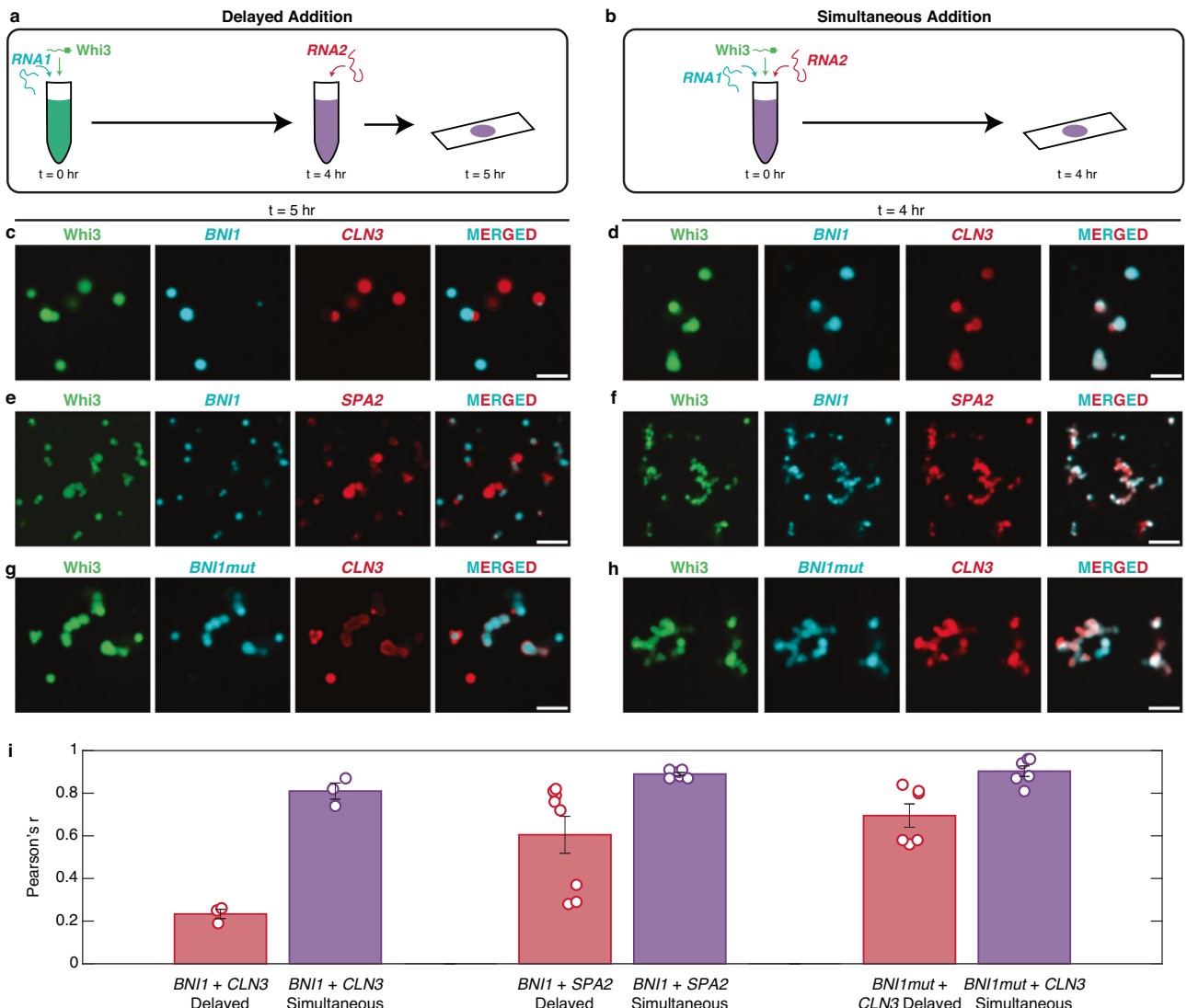

**Fig. 7 | Demixing of condensates in ternary systems is enabled by dynamical arrest caused by metastable traps in binary systems. a, b** Schematics of the mixing methods of Whi3 with two of its cognate RNAs. Whi3 and RNA concentrations are 5 μM and 5 nM, respectively. **c, d** Confocal images of condensates formed by Whi3, *BNI1*, and *CLN3* in **c** *delayed* versus **d** *simultaneous* modes. **e, f** Confocal images of condensates formed by Whi3, *BNI1*, and *SPA2* in **e** *delayed* versus **f** *simultaneous* modes. **g, h** Confocal images of condensates formed by Whi3,

*BNI1mut*, and *CLN3* in **g** *delayed* versus **h** *simultaneous* addition. Scale bar represents 5 μm for all images. **i** Pearson's *r*-values quantifying the colocalization of pairs of RNA molecules. Data for the *delayed* mode are shown in red, whereas data for the *simultaneous* mode are shown in purple. Error bars denote the standard error of mean across three, three, eight, six, six and six fields of view (circles) for each of the systems shown, respectively. Source data are provided as a Source Data file.

homotypic RNA-RNA interactions. Instead, the only mode available to *CLN3* is Whi3-*CLN3* interactions. We refer to this as a valence-limited mode of dynamical arrest that is in line with proposals made by Wingreen and coworkers[42]. Finally, while *BNI1* and *BNI1mut* have similar numbers of complementary sites for homotypic interactions (Supplementary Fig. 12), *BNI1mut* shows a lesser extent of demixing, thus highlighting the contribution of Whi3 cognate-binding sites to dynamically controlled demixing. This points to the role of site-specific interactions promoting dynamical arrest, a mode of arrest we refer to as specificity-limited dynamical arrest that is in line with observations made by Zhang and coworkers[46].

**Dynamical control over demixing of condensates is evident in cells**

Next, we asked if the demixing of Whi3-*CLN3* and Whi3-*BNI1* condensates is likely to be under dynamical control in live cells of *A. gossypii*. To test this possibility, we engineered an *A. gossypii CLN3*⁺/*BNI1*⁺

strain co-expressing the two transcripts under the same bidirectional promoter that should yield synchronous expression. Co-expression mimics the *simultaneous* mode that we deployed in vitro. We should see enhanced colocalization of the two transcripts if the timing of expression contributes to the extent of colocalization. A bidirectional promoter derived from the *A. gossypii* H2A/B histone locus was chosen to ensure robust co-expression in a wild-type background.

Consistent with previous results[8], wildtype expression showed a lack of colocalization between *CLN3* and *BNI1* (Fig. 9a, b). Furthermore, in the rare occurrences when *CLN3* and *BNI1* are colocalized, the degree of overlap is small, suggesting that these instances may be demixed condensates that are in spatial proximity to one another. In contrast, when *CLN3* and *BNI1* were co-expressed from the same promoter, the extent of colocalization of the two RNAs increased significantly (Fig. 9a, b). Pixel shift controls suggest that the increase in colocalization is not due to a high density of *CLN3* and *BNI1* in the *CLN3*⁺/*BNI1*⁺ co-expression strain (Supplementary Fig. 16a). We also

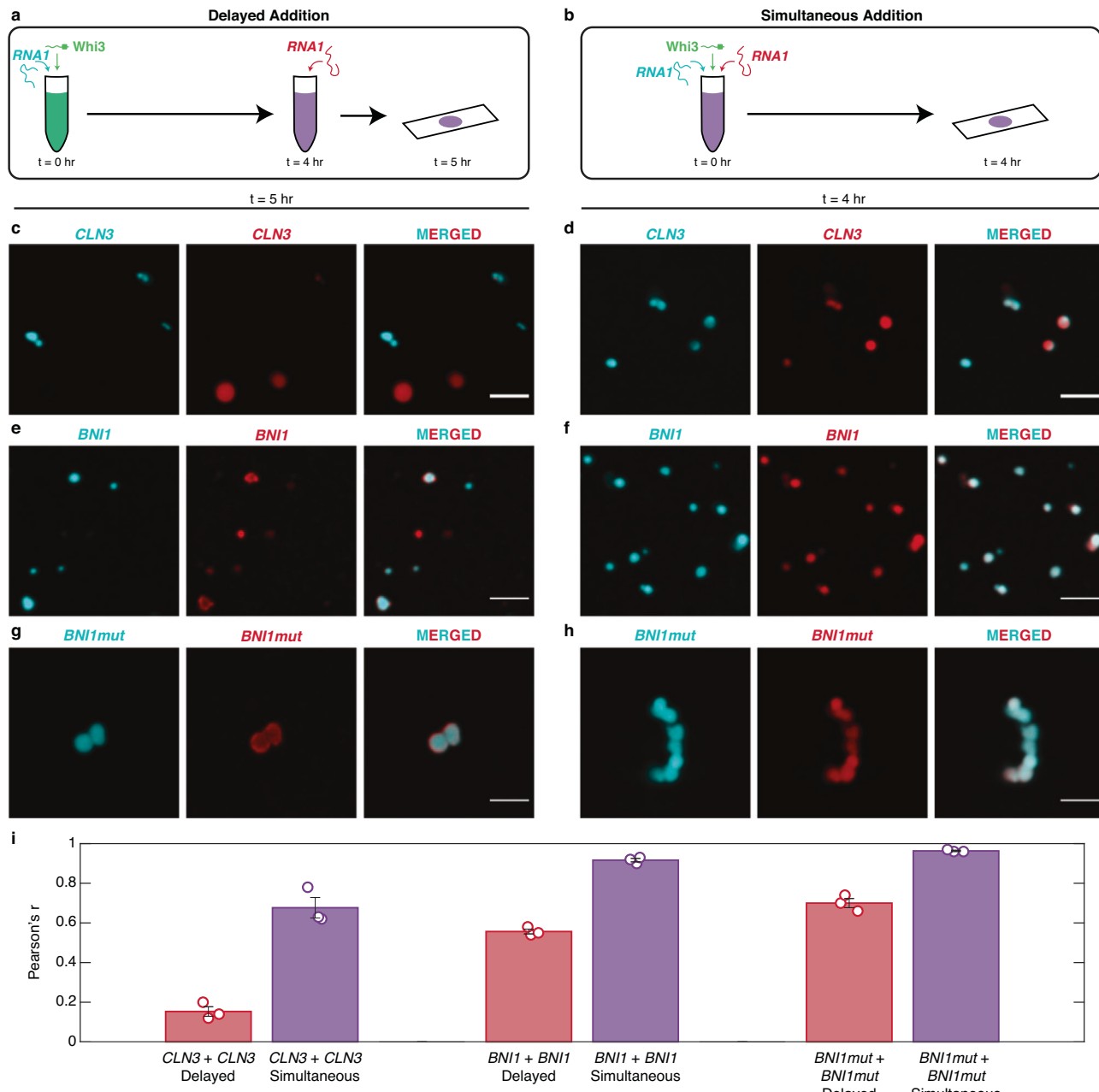

**Fig. 8 | Demixing is enabled by dynamical arrest even when the same RNA is added upon delay to the system. a, b** Schematics of the mixing methods of Whi3 with one of its cognate RNAs labeled with different dyes. Whi3 and RNA concentrations are 5 μM and 5 nM, respectively. **c, d** Confocal images of Whi3 with differently labeled *CLN3* added via delayed addition (**c**) or simultaneous addition (**d**). **e, f** Confocal images of Whi3 with differently labeled *BNI1* added via delayed addition (**e**) or simultaneous addition (**f**). **g, h** Confocal images of Whi3 with differently labeled *BNI1mut* added via delayed addition (**g**) or simultaneous addition (**h**). Scale bar represents 5 μm for all images. **i** Pearson's *r*-values of the colocalization of the differently labeled RNAs. Delayed addition is shown in red bars, whereas simultaneous addition is shown in purple bars. Error bars denote the standard error of mean across three independent fields of view (circles) for each of the systems shown. Source data are provided as a Source Data file.

observed a spatial component to the patterns of demixing versus colocalization. *A. gossypii* is an exemplar of multinucleated cells, and colocalized transcripts were more prominent near nuclei, where transcripts are synthesized (Supplementary Fig. 16b). Additionally, and intriguingly, when *CLN3* and *BNI1* were co-expressed from the same promoter, we observed polarity defects in *A. gossypii* cells. Specifically, synchronizing the expression of *CLN3* and *BNI1* resulted in impaired cell polarity as quantified by shorter hyphae lengths than what we observed in wild-type cells (Fig. 9c). These data suggest that precise spatiotemporal control of *CLN3* and *BNI1* transcripts is necessary for proper cell growth.

## Discussion

In this work, we investigated the determinants of compositionally distinct condensates formed by ternary mixtures comprising a shared component, Whi3, and different RNA molecules. Our findings point to significant contributions from dynamical control over compositional identities of condensates. The picture that emerges is of heterotypic interactions driving condensation, and dynamical arrest caused by a spectrum of RNA-mediated interactions, or lack thereof in the case of *CLN3*, causing demixing of condensates in ternary mixtures. We introduce three modes of demixing driven by dynamical arrest, which we refer to as crosslinking-limited, valence-limited, and specificity-

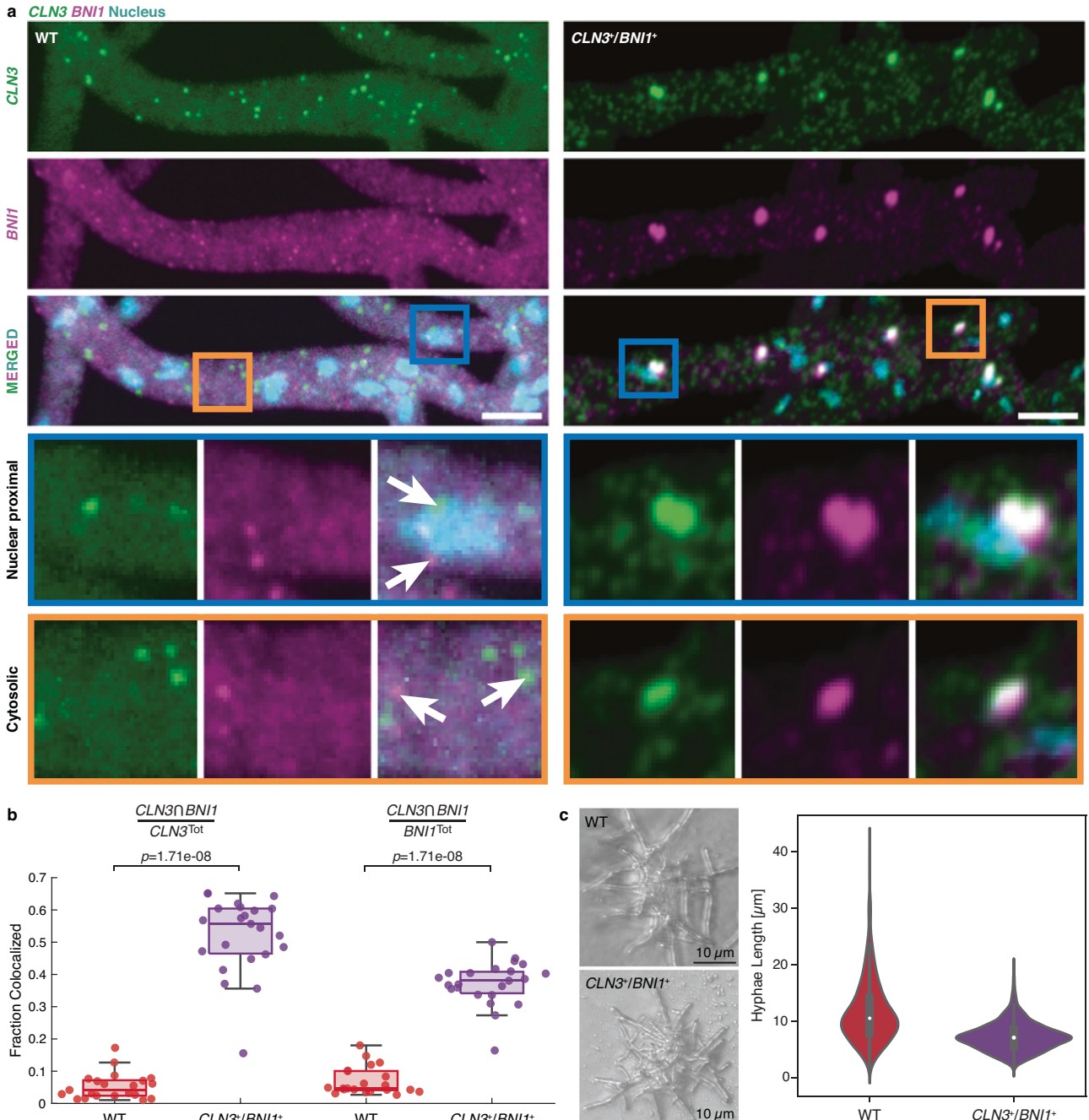

**Fig. 9 | Synchronized expression of *CLN3* and *BNI1* enables increased in colocalization in vivo. a** Representative images showing two-color smFISH of *CLN3* (green) and *BNI1* (magenta) in wild type (WT, left) and *CLN3⁺/BNI1⁺* (right) cells. Hoechst signals are shown in cyan, and the scale bar represents 5 μm. Insets show examples of nuclear proximal and cytosolic smFISH signals in the two strains. Arrows highlight individual *CLN3* and *BNI1* spots. **b** Quantification of the colocalizing fractions of *CLN3* and *BNI1* in WT (red) and *CLN3⁺/BNI1⁺* (purple) cells. Details of how colocalization was quantified are described in the methods. Each point denotes the fraction colocalized per image, *n*. Here, *n* = 21 and 23 for WT and *CLN3⁺/BNI1⁺*, respectively. For the boxplots, the median is shown as a colored horizontal line, whiskers show the 1.5 interquartile range, and the bottom and top of each box are the 25th and 75th percentiles, respectively. Any points beyond the whiskers are outliers. A two-sided Wilcoxon rank sum test was performed to extract *p*-values.

**c** Images from cells expressing 50 WT and 53 bidirectional mutant fungi were collected. Measurements were made on a total of 990 wild-type branches and 1261 mutant branches. ImageJ was used to measure branch lengths from the tip of each hypha to the closest branch point. Each measurement was used to plot violin plots in Seaborn for the wild-type and bidirectional mutant. For the boxplots, the white circle denotes the median, whiskers show the 1.5 interquartile range, and the bottom and top of each box are the 25th and 75th percentiles, respectively. We interpret shorter hyphae measurements in the bidirectional mutant as physiological defects that likely derive from the colocalization of *BNI1* and *CLN3* RNA. Images of fungi are representative and are intended to highlight the discrepancy in branch length between the wild-type and the bidirectional mutant. Source data are provided as a Source Data file.

limited, respectively. Importantly, our results show that enhanced colocalization of *CLN3* and *BNI1* in condensates with Whi3 can be realized by synchronizing their expression in vivo. This mimics the *simultaneous* mode of adding the three components in vitro. These experiments have biological significance because in wild-type cells, the levels, locations, and timings of expression of different RNAs are different. Therefore, we propose that the three modes of dynamical arrest might be important determinants of the demixing of Whi3-*BNI1* and Whi3-*CLN3* condensates in *A. gossypii*.

We propose that the role of dynamical arrest in patterning condensate composition may be quite general in directing condensate composition. Our findings are in line with those reported by Boeynaems et al.[47], who showed that condensates formed by simple peptide and RNA systems are characterized by exchange dynamics that occur on very different timescales. While peptides showed rapid recovery of FRAP, the corresponding timescales for RNA molecules that belong to the same condensates are considerably slower. Boeynaems et al. also showed that irregular condensate morphologies are a direct result of dynamical arrest caused by interactions among RNA molecules[47]. Similar results were reported by Ma et al.[48] and Seim et al.[5]. Going beyond RNA systems, the influence of dynamical arrest and different timescales for FRAP and different responses to mechanical forces has also been established for DNA systems as was shown by Keenen et al.[49]. Ranganathan and Shakhnovich showed that disparate timescales for the making and breaking of physical crosslinks versus the timescales of molecular transport inhibit the coarsening of dynamics[40]. Our results suggest that in addition to coarsening, one can also establish compositional identity through dynamical considerations. In this context, it is noteworthy that dynamical arrest can also give rise to dissipative structures[50] such as hollow condensates that form in active facsimiles of mitochondrial nucleoids[16].

There is growing interest in the determinants of compositionally distinct coexisting condensates[7,8,51] and the driving forces for forming multiphasic, multilayered condensates[16,52,53]. Proposals based on purely thermodynamic considerations focus on relative interfacial tensions[54], and hence relative solubilities, as the main determinants of condensate demixing or spatially organized structures of condensates[13,22,52–57]. Sequence-intrinsic[8,22,58,59] interactions, emulsification[60], chemical reactions[61], and elastic networks within cells[62,63] have been proposed as regulators of the sizes and compositional specificities of coexisting ribonucleoprotein condensates. Our work adds sequence-intrinsic, and composition-specific dynamical complexities of ribonucleoprotein mixtures to the list of possible controllers of compositional identities. Indeed, dynamical control cannot be ruled out in the observed "unblending" of transcriptional condensates that are influenced by repeat expansions that code for intrinsically disordered regions of transcription factors[51].

Our findings complement the recent work of Seim et al.[5], who titrated homotypic interactions among Whi3 proteins and found that this directly affects the Whi3:*CLN3* stoichiometry in condensates. Increasing the strengths of homotypic interactions among Whi3 proteins lowers protein concentrations within condensates, and slows the internal and exchange dynamics of molecules within Whi3:*CLN3* condensates. The effects of enhanced crosslinking among RNA molecules that lead to the dominance of percolation over phase separation was demonstrated using a Cahn-Hilliard-Cook system of equations[64] that modeled the coupling between gelation/percolation and phase separation dynamics. In the models of Seim et al.[5], dynamical arrest and suppression of coarsening via Ostwald ripening is a direct consequence of accelerated gelation (bond percolation) and long-lived physical crosslinks influenced by interactions among RNA molecules within condensates.

Overall, our findings bolster theoretical predictions of the thermodynamics of phase separation while also highlighting the relevance of long-lived crosslinks as a likely passive mechanism for demixing that

generates compositional identity in vitro and in cellulo[40,65]. The large cytoplasm of the filamentous fungus *A. gossypii*, and quite likely other mammalian cells such as axons[66], are likely candidates to leverage dynamical control to enable the formation of coexisting, demixed ribonucleoprotein condensates comprising the same protein and different RNA molecules.

## Methods

### LaSSI simulations
To investigate minimal models for associative polymers that can lead to equilibrium demixed condensates, we used a customized version of the lattice simulation engine LaSSI[15]. To generate plots and figures we used the common Python packages NumPy, Matplotlib and Seaborn. Adobe Illustrator was used to make the final figures.

### Simulations of ternary systems
For investigating the emergence of demixed condensates given two different RNA species with Whi3 protein, we simulated three different finite-sizes polymers. The Whi3 protein is modeled as a polymer containing three beads, $W_3$, while the two RNA species are modeled as eight bead polymers, $A_8$ and $B_8$, for *RNA1* and *RNA2* respectively. The beads are connected via implicit linkers with a length of two lattice sites. Using a lattice size of $L = 100$, we placed 5000 $W_3$, 1000 $A_8$, and 1000 $B_8$ molecules within the simulation boxes.

### Interaction models
To study scenarios that give rise to thermodynamical control of demixing of condensates generated by three-component systems that mimic the Whi3 protein + *RNA1* + *RNA2* systems, we used multiple interaction models to assess how different interactions among the components affect the overall phase behavior.

To set baseline expectations regarding the phase behaviors of three-component systems containing the Whi3 protein and two RNA species, we included heterotypic interactions between the Whi3 protein and the RNAs. This is referred to as the *Base Case*. Stickers interact via pairwise interactions, that are equivalent to the binding between the stickers. Given $\epsilon = -2k_B T$ as the energy scale, the interactions between the different stickers for the *Base Case* are defined as

$$\hat{\mathcal{E}}_{Base} = \begin{pmatrix} 0 & \epsilon & \epsilon \\ \epsilon & 0 & 0 \\ \epsilon & 0 & 0 \end{pmatrix}. \tag{1}$$

Here, the $\epsilon_{1,j}$ represents the interactions of the Whi3 protein sticker, $\epsilon_{2,j}$ and $\epsilon_{3,j}$ represent the interactions for *RNA1* and *RNA2*, respectively. Lastly, with this notation, $\epsilon_{i,i}$ represents homotypic interactions, and $\epsilon_{i,i \neq j}$ heterotypic interactions.

To test the effects of homotypic interactions between the RNAs, we added a scaling factor, $h$, and included homotypic interactions among the RNA stickers:

$$\hat{\mathcal{E}}_{Hom.Scaling} = \begin{pmatrix} 0 & \epsilon & \epsilon \\ \epsilon & h\epsilon & 0 \\ \epsilon & 0 & h\epsilon \end{pmatrix}, \tag{2}$$

where the scaling factor $h \in \{0, \frac{1}{2}, 1, \frac{3}{2}, 2\}$ determines the strength of the homotypic interaction for each RNA species with itself. The case where $h = 0$ represents the *Base Case* mentioned above, where no interactions occur between the RNAs.

To test whether asymmetric interactions between the Whi3 protein and an RNA species could lead to demixing, we applied a scaling

factor, $a$, to the heterotypic interactions in the *Base Case*:

$$\hat{\mathcal{E}}_{Asymm} = \begin{pmatrix} 0 & a\epsilon & \epsilon \\ a\epsilon & 0 & 0 \\ \epsilon & 0 & 0 \end{pmatrix}, \tag{3}$$

where the scaling factor $a \in \{\frac{1}{2}, 1, \frac{3}{2}, 2, 3\}$ determines the strength of the heterotypic interaction between Whi3 protein and *RNA1*. The case where $a = 1$ represents the *Base Case* where there are no differences between RNA species.

To set expectations about the phase behavior of a model that does lead to demixing, we added an additional isotropic repulsive interaction between the RNA species. The isotropic repulsion is a contact potential with a radius of one lattice site or $\sqrt{3}$ lattice units. Given $\epsilon_{rep} = 0.04 k_B T \approx \frac{1}{26} k_B T$, we have:

$$\hat{\mathcal{E}}_{Base} = \begin{pmatrix} 0 & \epsilon & \epsilon \\ \epsilon & 0 & 0 \\ \epsilon & 0 & 0 \end{pmatrix} \text{ and } \hat{\mathcal{E}}_{rep}^{ISO} = \begin{pmatrix} 0 & 0 & 0 \\ 0 & 0 & s\epsilon_{rep} \\ 0 & s\epsilon_{rep} & 0 \end{pmatrix}, \tag{4}$$

where the scaling factor $s \in \{1, 2, 5, 10\}$ determines the additional heterotypic isotropic repulsion between the different RNA species. Here, the RNAs do not repel their own species, but repel the other RNA species, which makes it less likely for different RNA species to be proximal. Again, if $s = 0$, we recover the *Base Case*.

### Simulation protocol
All simulations start with random initial conditions. For $t_{EQ} = 5 \times 10^7$ MC steps, the simulation temperature is set to $T_{EQ} = 10 \, T^*$. A constraining potential is applied to the system which pushes all molecules toward the center of the simulation box. This potential has the form:

$V(r, T) = T \, H(r - R_B) \, r^2$, where $r$ is the distance of a given bead from the center of the lattice, $R_B = 35$ lattice units, and $H(r)$ is the Heaviside function. Here, $T = T_{EQ}$ is a set constant. This potential resembles an indent-style potential implemented in LAMMPS[67]. All anisotropic/binding interactions are also turned off during this phase of the simulation. After $t_{EQ}$ MC steps, anisotropic/binding interactions are turned on and the temperature is exponentially annealed using an annealing protocol $T(t) = T_0 + T_{EQ} e^{-4\frac{t}{t_{EQ}}}$, to the target temperature of $T_0 = T^*$. As the temperature decreases and $T(t) - T_0$ gets lower than a threshold of 0.005, the temperature is set to $T_0$ and the biasing potential is turned off. The simulations are run for $t = 1 \times 10^{10}$ MC steps (see Supplementary Table 1 for details regarding the move set frequencies), and samples are only taken in the last half of each run. Samples are taken every $f_{data} = 2.5 \times 10^6$ MC steps, which result in 2000 samples for each simulation temperature. Three replicates per condition were used. The standard error of the mean between replicates is used as a measure of uncertainty.

### Generation of two-component phase diagrams with LaSSI
For generating the two-component phase boundaries, we sampled a large set of concentrations and stoichiometries and explicitly quantify the coexisting densities, when condensates are formed in the simulations. Since we have two explicit components, the system composition is determined by the numbers of each molecule and the overall concentration of the system. We fixed the total number of molecules to 5000 and changed the stoichiometry by changing the ratio between the two components. This results in 22 different stoichiometries. The simulation box size is then used to set the total concentration of the system. For each pair of molecule numbers, we have five different box sizes. Lastly, for each pair of molecule numbers, $(n_1, n_2)$, and box size $L$, we also sample the $(n_2, n_1)$ composition. This gives us a total of 220 independent compositions for a given interaction model. Details of the molecule numbers and move sets

for simulations of two-component systems may be found in Supplementary Tables 2 and 3.

All simulations start with random initial conditions. For $t_{EQ} = 5 \times 10^7$ MC steps, the simulation temperature is set to $T_{EQ} = 100 \, T^*$. A constraining potential is applied to the system which pushes all molecules toward the center of the simulation box. This potential has the form:

$V(r, T) = \Delta T \cdot r^2$, where $r$ is the distance of a given bead from the center of the lattice, $\Delta T$ is the temperature difference between the current simulation temperature and $T_0 = T^*$, the first target temperature. All anisotropic/binding interactions are turned off during these initial steps. After $t_{EQ}$ MC steps, anisotropic/binding interactions are turned on and the temperature is exponentially annealed as such $T(t) = T_0 + T_{EQ} e^{-4\frac{t}{t_{EQ}}}$, to the target temperature of $T_0 = T^*$. As the temperature decreases $\Delta T$ gets lower than a threshold of 0.005, the temperature is set to $T_0$, and the biasing potential is turned off. The simulations are run for $2 \times 10^9$ MC steps, and samples are only taken in the last half of each run. Samples are taken every $f_{data} = 1 \times 10^6$ MC steps, which result in 1000 samples for each run condition. Two replicates per condition were used.

### Analysis of data from LaSSI simulations
Given each simulation, we explicitly calculated the radial density profiles of each component from the center-of-mass (COM) of the system, and from the COM of the largest cluster of each component. The density profiles are generated by first computing a number histogram of beads, $H(r_n)$, from a given COM, with bin-width 0.25, from $r = 0$ to $r = \frac{\sqrt{3}L}{2}$, given a lattice size $L$. To normalize this number histogram, we explicitly calculated the number histogram of lattice sites, $H_0(r_n)$ for a given lattice size $L$ with periodic boundaries, and a bin-width of 0.25. Thus, the density profiles are calculated as:

$$\rho(r_n) = \frac{H(r_n)}{H_0(r_n)}. \tag{5}$$

To calculate the coexisting densities for the two-component systems, given a component $i$, we used the system COM to calculate the density profiles $\rho_i(r_n)$. We then averaged over the first 13 bins, and 20 bins near the end of the simulation box, avoiding the last 15 bins.

### Measure for demixing in three-component systems
Let component $i$ be the COM component, and let $j$ be the component for which we are calculating the density, then $\rho_{ij}(r_n)$ denotes the density profile of component $j$, given component $i$ as the COM. Given a density profile $\rho_{ij}(r)$, we can generate a normalized distribution, $\widetilde{\rho_{ij}}(r)$, such that:

$$\sum_{n=0}^{N_{bins}} \widetilde{\rho_{ij}}(r_n) = 1. \tag{6}$$

Then, using the Hellinger Distance[68], we can define a measure for demixing:

$$\mathcal{D}_{ij} = D_H(\widetilde{\rho_{ii}}, \widetilde{\rho_{ij}}), \tag{7}$$

where the Hellinger Distance, $D_H(P, Q)$, given distributions $P(x)$ and $Q(x)$, is defined as

$$D_H = \sqrt{1 - BC(P, Q)}, \tag{8}$$

and $BC$, the Bhattacharyya Coefficient, is defined as

$$BC(P, Q) = \sum_{x \in \mathcal{X}} \sqrt{P(x)Q(x)}. \tag{9}$$

Combining all definitions,

$$\mathcal{D}_{ij} = \sqrt{1 - \sum_{n=0}^{N_{bins}} \sqrt{\widetilde{\rho_{ii}}(r_n)\widetilde{\rho_{ij}}(r_n)}}. \qquad (10)$$

Thus, $\mathcal{D}_{ij}$ acts as a measure for the demixing between components $i$ and $j$.

Phase diagrams were plotted in terms of sticker concentration. The dilute phase arm of the phase diagram (x, y) was then mirrored by also plotting (y, x). The points that defined the dilute phase arm and the mirrored data ([x y], [y x]) were then fit to an ellipse using the guaranteed ellipse fitting method of Szpak et al.[32], and the mean square of the residuals $R^2$ of the fit was extracted.

## Protein purification and tagging

Full-length Whi3, with a N-terminal 6x His tag and TEV cleavage site, in BL21 cells from NEB (#C2527H) in TB media (Terrific Broth, Sigma #T0918-1KG) was induced with IPTG (Santa Cruz Biotechnology SC-202185B) to a final concentration of 1 mM at an $OD_{600}$ of 0.6–0.7 before being expressed overnight at 18 °C. These cells were then lysed in lysis buffer (1.5 M KCl [Sigma P9541-1KG], 50 mM HEPES pH 8.0 [Sigma 54457-50G-F], 20 mM Imidazole [Sigma I2399-500G], 5 mM BME [Sigma 444203-250 ML]) containing 10 mg of lysozyme (Sigma 9001-63-2), one Roche cOmplete™ protease inhibitor cocktail tablet (Sigma 5056489001), and PMSF (Sigma 10837091001) at a final concentration of 200 μM. The resulting lysate was sonicated on ice with a Branson SFX550 at 30% strength alternating 1 s on, 2 s off for 1 min. This was repeated five times, swirling the lysate gently between each sonication. The lysate was spun down and the supernatant passed over a HisTRAP FF column (Cytiva) on an ÄKTA pure 25 L (GE). The bound protein was eluted in 150 mM KCl, 50 mM HEPES pH 8.0, 200 mM Imidazole, and 5 mM BME. The eluate was then cleaved with 200 μg of TEV [pRK793 was a gift from David Waugh (Addgene plasmid # 8827; http://n2t.net/addgene:8827; RRID:Addgene_8827)]. The cleaved supernatant was concentrated in 3 kDa Amicon® centrifugal filter units (Millipore Sigma, UFC900324), and then injected onto a HiLoad 16/600 Superdex 200 pg column (Cytiva). Untagged Whi3 was then dialyzed into storage buffer (200 mM KCl, 50 mM Tris pH 8.0 [648311-1KG], 5 mM BME) and concentrated using 3 kDa Amicon® centrifugal filter units. For tagged Whi3, Alexa Fluor 488 NHS Ester (ThermoFisher, A20000) in DMSO (ThermoFisher D12345) was added at a ratio of 4:1 and incubated at room temperature with continuous mixing for 1 h in the absence of light. The tagged Whi3 was then loaded onto a HiLoad 16/600 Superdex 200 pg column with storage buffer, and the fractions were concentrated with 3 kDa Amicon® centrifugal filter units.

## RNA production, purification, and tagging

Plasmids in which the T7 promoter sequence was placed upstream of the coding regions for *CLN3*, *BNI1*, and *SPA2* were linearized with restriction enzymes to obtain a linearized template. *CLN3* RNA was transcribed with HiScribe™ T7 Quick High Yield RNA Synthesis Kit (NEB). *BNI1* and *SPA2* RNA were transcribed using Hi-T7® RNA Polymerase and Reaction Buffer (NEB) in lieu of T7 polymerase and buffer. For labeled RNA, 0.1 μL of 5 mM Cy3-UTP or Cy5-UTP (Cytiva, GEPA53026, GEPA55026) was added to each transcription reaction. After transcription, each reaction was treated with DNase before being purified with Monarch® RNA Cleanup Kit (NEB).

## In vitro measurements of phase boundaries

384-well plates with #1.5 polymer coverslip bottoms (Ibidi) were passivated for 15 min with 0.1% Tween-20 before being rinsed thrice with droplet buffer (150 mM KCl, 50 mM Tris pH 8.0, 5 mM BME). Untagged Whi3 protein and tagged RNA were diluted in droplet buffer and mixed to obtain the desired final concentration of protein and RNA. After incubation at room temperature for 1 h, the samples were visualized on a Zeiss Axiovert 200 M with a C-Apochromat 40×1.2NA water objective.

## Extracting the RNA apparent sticker valence from in vitro phase diagrams

To determine the overlap area, the MATLAB function *boundary* was used to define the boundaries of ($x_N$, $sy_N$) and ([$x_P$ $sy_P$], [$sy_P$ $x_P$]). Here, $x_N$ and $y_N$ define the x and y vectors of points along the upper boundary of the one-phase regime, $x_P$ and $y_P$ define the x and y vectors of points along the lower boundary of the two-phase regime, and $s$ is the scaling factor for the RNA sticker concentration. Then, the MATLAB function *polyshape* was used to create polygons defined by the boundaries. Next, we extracted the intersection of the two polygons using the MATLAB function *intersect* and calculated the area of this overlap region using the MATLAB function *area*. Then, the apparent sticker valence of the RNA was taken to be the $s$ value that yields the minimum overlap area. The boxplots in Fig. 4b correspond to the range in apparent sticker valence values if we allow for a five percent change in the minimum area.

## In vitro colocalization imaging

384-well plates with #1.5 glass coverslip bottoms (Cellvis) were passivated for 15 min with 0.1% Tween-20 before being rinsed thrice with droplet buffer (150 mM KCl, 50 mM Tris pH 8.0, 5 mM BME). For each sample, protein and RNA were diluted to 5 μM and 5 nM, respectively, in droplet buffer. For simultaneous samples, tagged Whi3 and tagged RNAs were added in quick succession to a low protein binding microcentrifuge tube (ThermoFisher) and allowed to incubate in darkness at room temperature for 4 h before being transferred to a passivated well and imaged on a Nikon Eclipse Ti2 equipped with a Yokogawa CSU-X1 Spinning Disk, 60X oil objective using Nikon Type F oil, and a Hamamatsu ORCA-Flash4.0 V3 camera. For delayed samples, tagged Whi3 and one of two tagged RNAs were mixed in a low protein binding microcentrifuge tube and allowed to incubate in darkness at room temperature for 4 h. Then the second RNA was added and allowed to incubate for 1 h before being transferred to a passivated well and imaged on the system described above.

## In vitro colocalization analysis

Images from three random fields of view for each sample were collected and cropped to 40,000 px² (approximately 469 μm²) in Fiji[69] The images were then split into separate channels and analyzed with the Coloc2 plugin, which assesses colocalization between two channels on a pixel-by-pixel basis, to obtain the Pearson correlation coefficient (Pearson's r). We employed a bisection threshold regression and 1000 randomizations with a PSF of 3.0. The Pearson's r was taken for each of the three fields of view and plotted with standard error.

## Fluorescence recovery after photobleaching (FRAP)

384-well plates with #1.5 glass coverslip bottoms (Cellvis) were passivated for 15 min with 0.1% Tween-20 before being rinsed thrice with droplet buffer (150 mM KCl, 50 mM Tris pH 8.0, 5 mM BME). Either tagged Whi3 or RNA is mixed with untagged RNA or Whi3 respectively and allowed to incubate for 1 h at room temperature. Samples were imaged on a Nikon A1Rsi with a 60X oil objective using Cargille Type B oil. Laser strength for photobleaching was 50% of the maximum intensity of the wavelength corresponding to the fluorophore used. Photobleaching occurred for 5 s with recovery measured for 3 min utilizing the laser corresponding to the fluorophore used.

The Region of Interest (ROI) and Time Analysis tools in Nikon Imagine Software (NIS) were used to trace the fluorescence intensity of three regions—the photobleached spot, an unbleached condensate, and a region with no condensates (i.e., background). Intensity changes over time were evaluated for all three regions and only acquisitions wherein the intensity of the unbleached condensate deviated less than

5% were used for analysis. Raw intensity values of the photobleached spots from three acquisitions were averaged, subtracted from the background, and normalized to the maximum average value. The "max" recovery value corresponds to the maximum value among all the recovery values (omitting the pre-bleach value). The standard error of the mean (SEM) was calculated for individual time points across the three acquisitions and for the maximum intensity values across three acquisitions. SEM was calculated by taking the standard deviation of time-point-matched intensity values or maximum intensity values and dividing this value by the square root of three (i.e., the $n$ of acquisitions). FRAP traces were generated in matplotlib.

### Complementary site analysis

GUUGle (https://bibiserv.cebitec.uni-bielefeld.de/guugle) was utilized to identify complementary sites between pairs of RNAs. Here, exact matches of length 11 or longer were identified using G-C, A-U, and G-U base pairing. Additionally, for the CLN3-BNI1, BNI1-SPA2, CLN3-CLN3, and BNI1-BNI1 pairs the mean SHAPE value was calculated over all nucleotides in the complementary sequences using the data from Langdon et al.[8].

### smFISH (single molecule Fluorescence In Situ Hybridization) and microscopy

RNA smFISH was performed as previously described[10]. Specifically, *Ashbya* cells were grown by inoculating dirty spores in 50 mL *Ashbya* full medium (AFM), for wild type or AFM supplemented with G418 (200 μg/mL) for the strain overexpressing *CLN3* and *BNI1* in a 500 mL baffled flask. Cells were grown shaking at 110 rpm at 30 °C for 16 h. After formaldehyde fixation and ethanol permeabilization, cells were probed with custom FISH probes from Stellaris. TAMRA-labeled probes against agCLN3 and Quasar-670-labeled probes against agBNI1 were both used at a final concentration of 2.5 nM and hybridized simultaneously at 37 °C overnight. Nuclei were stained with 5 μg/mL Hoechst in Wash Buffer (2x SSC, 10% v/v deionized formamide). Then, 20 μL of cells were then mounted in a 20 μL Vectashield mounting medium (Vector Laboratories, H-1000-10), sealed with a coverslip and imaged.

Imaging was performed on a Nikon Ti-E stand using a Yokogawa CSU-W1 spinning disk confocal unit. Images were acquired on a Plan-Apochromat 60x/1.49NA oil-immersion objective using a Zyla sCMOS camera (Andor) on Nikon NIS-Elements software v.4.60.

### In vivo *CLN3* and *BNI1* colocalization analysis

*CLN3* and *BNI1* spots were identified using the BIGFish Spot detection algorithm[70] that treats puncta as local maxima in the smFISH channels given a specified object radius. Puncta (spot) detection can be performed using two methods: (1) a single function call that enumerates the spots directly, or (2) a series of intermediate detection steps that yields the same results as in method 1, but with more debugging information. The latter option was chosen as it provided more data for troubleshooting the development of the spot detection pipeline. Using the latter method for detecting spots, a spot radius, $r$, of 150 nm (1.389 pixels) was used for detection. Maximum intensity projections were used, and images with uneven illumination and a few regions of non-overlapping hyphae were discarded. For each image in which the number of *CLN3* and *BNI1* spots were within an order of magnitude, the centers of identified RNA spots were collected. Then, a *CLN3* spot was determined to be colocalized with a *BNI1* spot if any *BNI1* center was a distance less than $2r$ away from the *CLN3* center. Fraction colocalized then refers to the number of all *CLN3* spots that had at least one *BNI1* spot less than $2r$ away divided by the total number of *CLN3* spots. For the pixel shift analysis, the centers of *CLN3* spots were shifted simultaneously by $2r$ in both the x and y directions before determining overlap. Then, using the shifted *CLN3* spot centers we determined if any *BNI1* spot was less than $2r$ away.

Proximity to the nucleus was determined by first identifying nuclei using the Cellpose 2.0 Python package[71]. Specifically, we ascribed a minimum object diameter of 10 voxels or 1.08 μm for the detection of regions of interest and performed segmentation. Larger object diameters perform poorly and detect false positives. Then, the centroids and areas of the nuclei masks were extracted using the Scikit-Image Python package[72]. Using the collected centroids and areas, we then determined whether the center of a *CLN3* spot was less than $r + R$ away from the center of any nucleus. Here, $R$ is the nucleus radius in pixels and was estimated to be $\sqrt{(A/\pi)}$, where $A$ is the area of the nucleus. *CLN3* spots less than $r + R$ away from the center of any nucleus were defined to be nuclear proximal and *CLN3* spots greater than or equal to $r + R$ were defined to be not nuclear proximal. Using these classifications, we then determined the fraction of *CLN3* spots colocalized with *BNI1* split by nuclear proximal and not nuclear proximal where colocalization was defined as any *BNI1* spot center less than $2r$ away from the *CLN3* center. The same analyses were also performed using *BNI1* as the reference RNA.

### Reporting summary
Further information on research design is available in the Nature Portfolio Reporting Summary linked to this article.

## Data availability
Raw experimental images, phase separation data, fluorescence recovery after photobleaching, and additional data needed to reproduce figures in this manuscript have been deposited in the Source Data file and/or at https://github.com/Pappulab/dynamical-control, which is publicly available as of the date of publication and has been archived in Zenodo with the identifier https://doi.org/10.5281/zenodo.1004269. Protein and RNA expression constructs for in vitro measurements are available upon request. Correspondence and requests for materials should be addressed to A.S.G. and R.V.P. Source data are provided with this paper.

## Code availability
The latest, public release of LaSSI, developed and maintained by the Pappu lab, is available on the Pappu lab GitHub repository. Please visit https://github.com/Pappulab/LASSI. Specific analysis routines, the simulation results, and the code used to analyze in vitro as well as in vivo data are available at https://github.com/Pappulab/dynamical-control and have been archived in Zenodo with the identifier https://doi.org/10.5281/zenodo.1004269.

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

## Acknowledgements

This work was supported by grants from the Air Force Office of Scientific Research (grant FA9550-20-1-0241 to A.S.G. and R.V.P.), the St. Jude Research Collaborative on the Biology and Biophysics of RNP granules (to R.V.P.), and the National Institutes of Health (F32GM146418-01A1 to M.R.K. and R01NS121114 to R.V.P.). The contributions of J.M.L., A.E.P., and N.A.E. were supported, in part, by the Center for Biomolecular Condensates in the James F. McKelvey School of Engineering at Washington University in St. Louis.

## Author contributions

A.S.G. and R.V.P. came up with the project idea. A.Z.L. performed all the in vitro experiments and generated all the constructs with the help of M.R.K. and A.E.P. K.M.R. and F.D. performed the LaSSI simulations and developed the theoretical framework to extract insights regarding shapes of phase boundaries. A.Z.L., K.M.R., F.D., A.E.P. and R.V.P. worked together to analyze and interpret the in vitro data. M.R.K. helped with generating constructs, the FRAP measurements, and with troubleshooting microscopy measurements. A.J. and A.S.G. designed the cellular measurements, and A.J. performed the cellular measurements. J.M.L. provided technical expertise to analyze in vivo data. N.A.E. performed experiments to measure protein-free condensation. I.S. provided independent assessments of some of the in vitro measurements. A.Z.L., K.M.R., F.D. and R.V.P. were the main authors of all but the penultimate drafts of the manuscript. All authors read and edited the final manuscript.

## Competing interests

R.V.P. is a member of the scientific advisory board and a shareholder of Dewpoint Therapeutics. The work reported here was not influenced by this affiliation. The remaining authors declare no competing interests.
