## [Peer review file · Nature Communications]

REVIEWER COMMENTS

Reviewer #1 (Remarks to the Author):

The authors investigated the formation of demixed biomolecular condensates containing two different RNA types and a common protein. Via a combination of computer simulations, and in vitro and in vivo experiments, they demonstrated that demixing is not driven by equilibrium thermodynamics but depends on the history of how biomolecular condensates were formed due to strong RNA-RNA and protein-RNA interactions. In vitro experiments demonstrated that demixing is more likely when two different RNA types are introduced sequentially while mixing tends to occur when both RNA types are introduced simultaneously. Remarkably, this was confirmed with in vivo experiments with an engineered *A. gossypii* strain, which used a common promoter to transcribe two different types of RNA and demonstrated enhanced colocalization of different RNAs compared to the experiments with a wild-type strain, where the production of RNAs was asynchronous. The manuscript is well-written and would be of interest to a broad spectrum of readers. However, the manuscript would benefit by addressing the following comments.

1.) Authors inferred the relative contribution of homotypic and heterotypic interactions in Whi3-RNA mixtures via comparisons with computer simulations. While the simulations were in thermal equilibrium, demixed phases in in vitro Whi3-RNA mixtures showed dynamical arrests. The authors should comment on whether it is safe to assume that the measured phase boundaries in Fig. 3 correspond to the equilibrium thermodynamics and that the dynamical arrest only occurs deep in the phase separation region.

2.) For the in vivo experiments with the wild-type strain in Fig. 9a, it seems that all CLN3 and BNI1 condensates are separated in space. Are there any examples, where CLN3 and BNI1 condensates are sticking together in demixed states similar to in vitro experiments in Fig. 7c? This would further strengthen the claim that synchronous production in both space and time is required for the production of demixed condensates in vivo.

Minor comments:

3.) In captions of Fig. 1, explain that the numbers $0 \times \epsilon_{\text{het}}$ and $1.5 \times \epsilon_{\text{het}}$ on panels b and c, respectively correspond to the magnitude of homotypic interactions.

4.) In Fig. 2b and Fig. 3b denote the concentrations with [Whi3 Stickers] and [RNA Stickers] like it was done in Fig. 2c-h, Fig. S8, and Fig. S9.

5.) In captions of Fig. 4, clarify that the Hoechst signals in cyan color are used to stain the nuclei.

6.) Figures S11 and S12 are referenced before Fig. S10 in the main text.

7.) Figures S16 is referenced before Fig. S15 in the main text.

8.) What was the purpose of averaging densities over the first 13 bins and 20 bins near the end of the simulation box?

9.) Remove empty page S3 in the supplemental materials.

10.) Typos:

- line 57: "analysesimplicate"
- line 146: "facto. ."
- line 462: "supplemental materials" should be "methods"
- line 587: t should be t_{EQ}
- line 1080: "supplemental materials" should be "methods"

Reviewer #2 (Remarks to the Author):

The manuscript by Lin et al. has improved substantially since the initial submission. The addition of new experimental data, e.g., in Figure 5, makes for a more complete story. In a crucial change, the revised manuscript no longer claims that it is unphysical for heterotypic interactions between various RNAs to differ from homotypic RNA interactions. The revised manuscript also more carefully notes caveats with respect to the interpretation of the in vitro data.

Overall, the manuscript shows that dynamical considerations play a dominant role over thermodynamics in this system (regardless of what the thermodynamically favored state might be) due to the low mobilities of RNAs in condensates. This conclusion is well supported by the experimental data. Along these lines, I suggest that the authors consider revising the following sentence in the abstract: "Consequently, in ternary systems, compositionally distinct Whi3-RNA condensates arise due to dynamical arrest driven by cognate Whi3-RNA interactions and intra-condensate homotypic RNA interactions." As written, this sentence could be interpreted as claiming that the compositions are determined solely by dynamics, whereas the data support the slightly weaker conclusion that dynamical arrest is "a player in controlling demixing and the generation of compositionally distinct condensates", as the authors write in their letter.

I still do not believe that it is clear from the in vitro data what the thermodynamic ground state is. The authors argue that "Our data clearly point to condensates being significantly more well-mixed with simultaneous addition, indicating the more well-mixed assembly is likely a thermodynamic ground state." However, it is also possible that dynamical arrest prevents equilibration on the experimental timescale: Since it is clear that the RNA molecules have low mobilities in the condensates, it is possible that different RNAs that randomly end up in the same condensate upon simultaneous mixing simply do not have enough time to demix. This issue was also brought up by the third reviewer in *Comment 7. While the authors have clearly explained their image analysis approach in their response, the data in Figure 7d-h nonetheless do not appear to be as well mixed (at least by eye) as one would expect if RNA mixing were favored thermodynamically. Note that this issue does not detract from the main conclusion of the paper, namely that dynamical arrest is a key determinant of condensate compositions on physiologically relevant timescales.

There is a typo line 57.

Responses to reviewers

Responses to comments of Reviewer 1

Comment 1: *The authors investigated the formation of demixed biomolecular condensates containing two different RNA types and a common protein. Via a combination of computer simulations, and in vitro and in vivo experiments, they demonstrated that demixing is not driven by equilibrium thermodynamics but depends on the history of how biomolecular condensates were formed due to strong RNA-RNA and protein-RNA interactions. In vitro experiments demonstrated that demixing is more likely when two different RNA types are introduced sequentially while mixing tends to occur when both RNA types are introduced simultaneously. Remarkably, this was confirmed with in vivo experiments with an engineered *A. gossypii* strain, which used a common promoter to transcribe two different types of RNA and demonstrated enhanced colocalization of different RNAs compared to the experiments with a wild-type strain, where the production of RNAs was asynchronous. The manuscript is well-written and would be of interest to a broad spectrum of readers. However, the manuscript would benefit by addressing the following comments.*

Response to comment 1: We thank the reviewer for their assessment of our work and have addressed their comments below.

Comment 2: *Authors inferred the relative contribution of homotypic and heterotypic interactions in *Whi3*-RNA mixtures via comparisons with computer simulations. While the simulations were in thermal equilibrium, demixed phases in in vitro *Whi3*-RNA mixtures showed dynamical arrests. The authors should comment on whether it is safe to assume that the measured phase boundaries in Fig. 3 correspond to the equilibrium thermodynamics and that the dynamical arrest only occurs deep in the phase separation region.*

Response to comment 2: The measured phase boundaries delineate concentration regimes where phase separation is observed versus regions where phase separation is not observed. It appears that considerations of dynamical arrest do not influence the onset of phase separation, which we use to delineate phase boundaries. They will however impact studies of condensate dissolution, which are not relevant for the part where we focus on the mapping of phase boundaries to infer the balance between homotypic and heterotypic interactions as drivers of phase separation.

Comment 3: *For the in vivo experiments with the wild-type strain in Fig. 9a, it seems that all *CLN3* and *BN11* condensates are separated in space. Are there any examples, where *CLN3* and *BN11* condensates are sticking together in demixed states similar to in vitro experiments in Fig. 7c? This would further strengthen the claim that synchronous production in both space and time is required for the production of demixed condensates in vivo.*

Response to comment 3: In our analysis of *in vivo* data, we find that the mutant has more overlap than wildtype. There are cases where *CLN3* and *BN11* condensates appear to be sticking together, at least within the 2-radius cutoff. In general, synchronized expression leads to higher degrees of overlap, whereas the wild-type expression leads to condensates that touch one another instead of overlapping (see Figure S15). In the revision, we have added the following sentence (see lines 450-452): “Furthermore, in the rare occurrences when *CLN3* and *BN11* are colocalized, the degree of overlap is small, suggesting that these instances may be demixed condensates that are in proximity (Fig. S15).”

Comment 4: *In captions of Fig. 1, explain that the numbers $0 \times \epsilon_{\text{het}}$ and $1.5 \times \epsilon_{\text{het}}$ on panels b and c, respectively correspond to the magnitude of homotypic interactions.*

Response to comment 4: We have added the following clarifications to the caption for Fig. 1. “Here, $\epsilon_{\text{Het}} = -2k_B T$ refers to the strength of RNA interactions with the Whi3 mimic and $\epsilon_{\text{Hom}} = 0 \times \epsilon_{\text{Het}}$ implies that homotypic interactions are zeroed out. (c) In contrast, demixed condensates are formed when the RNAs have strong homotypic interactions as well, where the strength of homotypic interactions is set to be 1.5 times that of heterotypic interactions, i.e., $\epsilon_{\text{Hom}} = 1.5 \times \epsilon_{\text{Het}}$.”

Comment 5: *In Fig. 2b and Fig. 3b denote the concentrations with [Whi3 Stickers] and [RNA Stickers] like it was done in Fig. 2c-h, Fig. S8, and Fig. S9.*

Response to comment 5: Fig. 2b and Fig. 3b have been updated with the requested change.

Comment 6: *In captions of Fig. 4, clarify that the Hoechst signals in cyan color are used to stain the nuclei.*

Response to comment 6: We believe the reviewer is referring to Fig. 9. The caption for this figure states the Hoechst signals are shown in cyan.

Comment 7: *Figures S11 and S12 are referenced before Fig. S10 in the main text.*

Response to comment 7: We thank o the reviewer for pointing this out. We have updated the ordering of our supplementary figures.

Comment 8: *Figures S16 is referenced before Fig. S15 in the main text.*

Response to comment 8: We thank the reviewer for pointing this out. We have updated the ordering of our supplementary figures.

Comment 9: *What was the purpose of averaging densities over the first 13 bins and 20 bins near the end of the simulation box?*

Response to comment 9: This is based on the simulation size and bin sizes. The first 13 will capture the dense phase density, the last 20 capture the dilute phase density. This was established as a means of calculating co-existing densities in LaSSI simulations e.g., Ruff et. al., 2021, *PNAS*.

Comment 10: *Remove empty page S3 in the supplemental materials.*

Response to comment 10: The empty page has been removed.

Comment 11:

Typos:

- line 57: “*analysesimplicate*”

- line 146: “*facto. .*”

- line 462: “*supplemental materials*” should be “*methods*”

- line 587: *t* should be *t_EQ*

- line 1080: “*supplemental materials*” should be “*methods*”

Response to comment 11: We thank the reviewer for catching these typos. We have corrected them.

Responses to comments of Reviewer 2

Comment 1: *The manuscript by Lin et al. has improved substantially since the initial submission. The addition of new experimental data, e.g., in Figure 5, makes for a more complete story. In a crucial change, the revised manuscript no longer claims that it is unphysical for heterotypic interactions between various RNAs to differ from homotypic RNA interactions. The revised manuscript also more carefully notes caveats with respect to the interpretation of the in vitro data.*

Response to comment 1: We thank the reviewer for their feedback and their positive comments regarding the revised version of the manuscript.

Comment 2: *Overall, the manuscript shows that dynamical considerations play a dominant role over thermodynamics in this system (regardless of what the thermodynamically favored state might be) due to the low mobilities of RNAs in condensates. This conclusion is well supported by the experimental data. Along these lines, I suggest that the authors consider revising the following sentence in the abstract: "Consequently, in ternary systems, compositionally distinct Whi3-RNA condensates arise due to dynamical arrest driven by cognate Whi3-RNA interactions and intra-condensate homotypic RNA interactions." As written, this sentence could be interpreted as claiming that the compositions are determined solely by dynamics, whereas the data support the slightly weaker conclusion that dynamical arrest is "a player in controlling demixing and the generation of compositionally distinct condensates", as the authors write in their letter.*

Response to comment 2: The journal now requires an abstract that has no more than 150 words. In the abridged abstract, we have taken care to tone down the statement referenced by the reviewer. The new abstract reads as follows: "Cellular matter can be organized into compositionally distinct biomolecular condensates. For example, in *Ashbya gossypii*, the RNA binding protein Whi3 forms distinct condensates with different RNA molecules. Using criteria derived from a physical framework for explaining how compositionally distinct condensates can form via purely thermodynamic considerations, we find that condensates in vitro form mainly via heterotypic interactions in binary mixtures of Whi3 and RNA. However, within these condensates, RNA molecules become dynamically arrested. As a result, in ternary systems, simultaneous additions of Whi3 and pairs of distinct RNA molecules lead to well-mixed condensates, whereas delayed addition of an RNA component results in compositional distinctness. **Therefore, compositional identities of condensates can be achieved via dynamical control, being driven, at least partially, by dynamical arrest of RNA molecules.** Finally, we show that synchronizing the production of different RNAs leads to more well-mixed, as opposed to compositionally distinct condensates in vivo." We draw the reviewer's attention to the sentence in bold face.

Comment 3: *I still do not believe that it is clear from the in vitro data what the thermodynamic ground state is. The authors argue that "Our data clearly point to condensates being significantly more well-mixed with simultaneous addition, indicating the more well-mixed assembly is likely a thermodynamic ground state." However, it is also possible that dynamical arrest prevents equilibration on the experimental timescale: Since it is clear that the RNA molecules have low mobilities in the condensates, it is possible that different RNAs that randomly end up in the same condensate upon simultaneous mixing simply do not have enough time to demix. This issue was also brought up by the third reviewer in *Comment 7. While the authors have clearly explained their image analysis approach in their response, the data in Figure 7d-h nonetheless do not appear to be as well mixed (at least by eye) as one would expect if RNA mixing were favored thermodynamically. Note that this issue does not detract from the main conclusion of the paper;*

namely that dynamical arrest is a key determinant of condensate compositions on physiologically relevant timescales.

Response to comment 3: We agree that we cannot definitively establish the thermodynamic ground state. Therefore, we have reworded the caption in Figure S14 to as follows: “We propose that the data shown here are suggestive of the well-mixed condensates being thermodynamic ground states. This proposal is based on two other observations, including the results of Langdon et al., who showed that preparing condensates following heat treatments of the RNA molecules of interest, leads to well-mixed condensates as opposed to demixed condensates. Similar results were reported by Boeynaems et al., in their study of ternary mixtures of arginine-rich peptides and different, base-pairing RNA molecules. Heating and annealing assays will drive unfolding of both the RNA and Whi3 RRM. The annealing protocol would have to be sufficiently slow to allow for refolding of the molecules and remodeling of the condensates. An optimal protocol for achieving this remains elusive. Therefore, for now, we propose, based on precedents in the literature, that well-mixed condensates are likely to be the thermodynamic ground states. A corollary of this proposal is that demixed, compositionally distinct condensates with Whi3 as the shared component are metastable.”

Comment 4: *There is a typo line 57.*

Response to comment 4: This and other typos have been fixed.